# Reconsidering Faithfulness in Regular, Self-Explainable, and Domain Invariant GNNs

**Steve Azzolin**[*]  **Antonio Longa**[*]  **Stefano Teso**[*]  **Andrea Passerini**[*]

## Abstract

As Graph Neural Networks (GNNs) become more pervasive, it becomes paramount to build reliable tools for explaining their predictions. A core desideratum is that explanations are *faithful*, *i.e.*, that they portray an accurate picture of the GNN's reasoning process. However, a number of different faithfulness metrics exist, begging the question of what is faithfulness exactly and how to achieve it. We make three key contributions. We begin by showing that *existing metrics are not interchangeable – i.e.*, explanations attaining high faithfulness according to one metric may be unfaithful according to others – and can *systematically ignore important properties of explanations*. We proceed to show that, surprisingly, *optimizing for faithfulness is not always a sensible design goal*. Specifically, we prove that for injective regular GNN architectures, perfectly faithful explanations are completely uninformative. This does not apply to modular GNNs, such as self-explainable and domain-invariant architectures, prompting us to study the relationship between architectural choices and faithfulness. Finally, we show that *faithfulness is tightly linked to out-of-distribution generalization*, in that simply ensuring that a GNN can correctly recognize the domain-invariant subgraph, as prescribed by the literature, does not guarantee that it is invariant unless this subgraph is also faithful. The code is publicly available on GitHub[1].

## 1 Introduction

The increasing popularity of GNNs (Scarselli et al., 2008; Kipf and Welling, 2016; Veličković et al., 2018) for even high-stakes tasks (Agarwal et al., 2023) has prompted the development of tools for explaining their decisions. Regular GNNs are opaque in that their decisions can only be explained in a *post-hoc* fashion using specialized tools (Longa et al., 2024), whereas self-explainable GNNs are designed to natively output explanations for their predictions (Kakkad et al., 2023). A key metric for evaluating explanations is **_faithfulness_** (Pope et al., 2019; Yuan et al., 2022; Amara et al., 2022; Zhao et al., 2023; Tan et al., 2022; Zheng et al., 2023). Intuitively, an explanation is faithful as long as it *highlights all and only those elements – edges and/or node features – of the input graph that are truly relevant for the prediction*. Faithful explanations, therefore, portray an accurate picture of the GNN's working and thus support understanding, trust allocation, and debugging (Teso et al., 2023).

Existing faithfulness metrics assess the stability of the model's output to perturbations of the input – *e.g.*, to ensure that modifying the elements marked as irrelevant by the explanation has in fact no effect – yet differ in many non-trivial details. For instance, some metrics perturb the input by zeroing out all irrelevant node features (Agarwal et al., 2023) while others delete a subset of edges (Zheng et al., 2023); see Section 3 for an overview. Unfortunately, the literature provides little guidance on selecting appropriate metrics and tuning GNN architectures for faithfulness.

**Not all metrics are the same.** (Section 3) We begin by parameterizing existing metrics along two dimensions: how *stability* is measured and what *perturbations* are allowed. We show that different parameter choices yield rather different metrics, in the sense that **_explanations that are faithful for one metric may not be faithful according to others_**. These differences are far reaching. In fact, we show that different metrics can **_rank explanation algorithms differently_** and that popular metrics can be **_systematically insensitive to the number of irrelevant elements captured by the explanation_**.

---

[*]University of Trento, {name.surname}@unitn.it

[1]https://github.com/steveazzolin/reconsidering-faithfulness-in-gnns

It follows that faithfulness measurements cannot be interpreted unless their parameters are known: this is relevant when evaluating XAI algorithm and in high-stakes applications, like loan approval, where explanation providers could manipulate (but withhold) the parameters to mislead end-users into overestimating the faithfulness of their explanations (Bordt et al., 2022).

**Is faithfulness always worth optimizing for?** (Section 4) Then, we study to what extent optimizing for faithfulness is a sensible design goal. We show that for GNNs satisfying injectivity, ***explanations achieving perfect faithfulness are not informative***, and identify a natural trade-off between expressiveness of the model and usefulness of faithful explanations. At the same time, we show that for self-explainable GNNs and domain invariant GNNs, which employ a modular architecture, strict faithful explanations *can* be informative, and evaluate how architectural design choices can impact faithfulness, highlighting how popular models implicitly penalize faithfulness.

**Faithfulness is key to OOD generalization.** (Section 5) Finally, we highlight the central but so far neglected role of faithfulness in *domain invariance*. Prior work (Cai et al., 2023; Chen et al., 2023; Gui et al., 2023; Jiang et al., 2023) tackles domain invariance by constructing modular GNNs that isolate the domain invariant portion of the input and use it for prediction. We show that ***extracting a domain invariant subgraph is not enough for a GNN to be truly domain invariant***: unless the subgraph is also faithful, the information from the domain-dependent components of the input can still influence the prediction, thus preventing domain invariance. This reveals a key limitation of current design and evaluation strategies for domain-invariant GNNs, which neglect faithfulness altogether.

Overall, our hope is that these contributions will prompt researchers to reconsider faithfulness in a broad sense, *i.e.*, to reconsider *i)* how it should be computed, *ii)* how GNN architectures should achieving it, and *iii)* its role outside of the XAI literature.

## 2   GRAPH NEURAL NETWORKS AND FAITHFULNESS

Throughout, we indicate graphs as $G = (V, E)$ and annotated graphs as $G_A = (G, X)$, where $\mathbf{x}_u \in \mathbb{R}^d$ are per-node features. We denote multisets as $\{\{\ldots\}\}$, the $k$-hop neighborhood of $u$ as $N_k(u)$, and shorten $N_1(u)$ to $N(u)$, and $\|G\| = |E|$ to $m$.

**Graph Neural Networks** (GNN) (Scarselli et al., 2008) are discriminative classifiers that, given an input graph $G_A$, define a conditional distribution $p_\theta(\cdot \mid G_A)$ over candidate labels. In graph classification, the label $y \in \{1, \ldots, c\}$ applies to the whole graph, while in node classification it is a vector $\mathbf{y} \in \{1, \ldots, c\}^n$ with one element per node. Inference in GNNs is *opaque*, in that it relies on message passing of embedding vectors along the graph's topology. Usually, this amounts to recursively applying an update-aggregate operation of the form:

$$\mathbf{h}_u^\ell = \mathsf{update}(\mathbf{h}_u^{\ell-1}, \mathsf{aggr}(\{\{\mathbf{h}_v^{\ell-1} \; : \; v \in N(u)\}\})) \tag{1}$$

to all nodes $u$, from the bottom to the top layer. Here, $\mathbf{h}_u^0 = \mathbf{x}_u$ are the node features, $\mathbf{h}_u^\ell$ the node embeddings at the $\ell$-th layer, update a learnable non-linear function, and aggr a permutation-invariant neighborhood aggregator (Bacciu et al., 2020; Bronstein et al., 2021). We refer to this form of aggregation as *local*, as opposed to *non-local* aggregation where aggr runs over nodes outside of $N(u)$, *e.g.*, via virtual nodes (Sestak et al., 2024). In node classification, the top-layer embeddings are stacked into a matrix $H^L \in \mathbb{R}^{n \times d}$, which is fed to a dense layer to obtain a label distribution $p_\theta(\mathbf{Y} \mid G_A) = \mathsf{softmax}(H^L W)$, where $W \in \mathbb{R}^{d \times c}$ are weights. In graph classification, they are aggregated into an overall embedding $\mathbf{h}_G = \mathsf{aggr}_G(\{\{\mathbf{h}_u^L : u \in V\}\})$, also fed to a dense layer.

**Explanations and Faithfulness.** There exist a number of post-hoc techniques (Longa et al., 2024; Agarwal et al., 2023; Kakkad et al., 2023) that given a GNN $p_\theta$ can extract a *local explanation* for any target decision $(G_A, \hat{y})$. These explanations identify a subgraph $R_A$ of $G_A$ capturing those elements – edges and/or features – deemed relevant for said decision. Unfortunately, explanations output by post-hoc approaches may fail to identify all and only the truly relevant elements,[2] in which case we say they are not *faithful* to the GNN's reasoning process (Longa et al., 2024; Agarwal et al., 2023). Lack of faithfulness hinders understanding, trust modulation, and debugging (Teso et al., 2023).

This has prompted the development of ***self-explainable GNNs*** (SE-GNNs), a class of GNNs that natively output explanations without any post-hoc analysis (Kakkad et al., 2023; Christiansen et al.,

---

[2]The tacit assumption in the XAI and domain invariance literature is that these elements exists.

2023). SE-GNNs comprise two modules: the *detector* $f$ extracts a class-discriminative subgraph $R_A$ from the input, and the *classifier* $g$ uses $R_A$, and only $R_A$, to infer a prediction. In this context, $R_A$ acts as a local explanation. Both modules are GNNs, and the detector is encouraged to output human interpretable explanations by leveraging attention (Miao et al., 2022a; Lin et al., 2020; Serra and Niepert, 2022; Wu et al., 2022), high-level concepts or prototypes (Zhang et al., 2022; Ragno et al., 2022; Dai and Wang, 2021; 2022; Magister et al., 2022), or other techniques (Yu et al., 2020; 2022; Miao et al., 2022b; Giunchiglia et al., 2022).

In practice, the detector outputs per-element relevance scores (usually in $[0, 1]$), which are then used to identify an explanation subgraph $R_A$ via, *e.g.*, thresholding or topK. Importantly, the node features $X^R \in \mathbb{R}^{|R_A| \times d}$ of $R_A$ are derived from, but *not necessarily identical to*, those of $G_A$. These remarks will become relevant when discussing strategies for improving faithfulness in Section 4.2.

**Domain Invariance.** *Domain-invariant GNNs* (DI-GNNs) aim to generalize across related domains (Cai et al., 2023; Chen et al., 2023; Gui et al., 2023; Jiang et al., 2023). The underlying assumption is that the label $y$ depends only on (hidden) *domain-invariant* factors, while the input graph $G_A$ is contaminated by domain-dependent elements. The literature suggests that, in order to generalize across domains, DI-GNNs have to be ***plausible***, that is, they should recover the (unobserved) truly invariant subgraph as well as possible (Schölkopf et al., 2021; Jiang et al., 2023).

Like SE-GNNs, DI-GNNs are modular, *i.e.*, they pair a detector $f$ for identifying the invariant subgraph $R_A$, which plays the role of an explanation, with a classifier $g$ taking $R_A$ as input. The detector is encouraged to output highly plausible subgraphs through specialized regularization terms and training strategies (Li et al., 2022; Chen et al., 2022; 2023; Gui et al., 2023). We will show in Section 5 that this is not enough to ensure the model as a whole is domain invariant.

# 3  ARE ALL FAITHFULNESS METRICS THE SAME?

In the remainder, we fix a GNN $p_\theta$ and a decision $(G_A, \hat{y})$[3] and study the faithfulness of a corresponding local explanation $R_A$. Intuitively, an explanation is faithful insofar as it is sufficient, *i.e.*, keeping it fixed shields the model's output from changes to its complement $C_A$, and necessary, *i.e.*, altering it affects the model's output even if the complement $C_A$ is kept fixed (Watson et al., 2021; Beckers, 2022; Marques-Silva and Ignatiev, 2022; Darwiche and Hirth, 2023).

**Definition 1.** *An explanation $R_A$ is **strictly sufficient** if no change to the complement $C_A$ does induce any change in the model's output, **strictly necessary** if all changes to the explanation do, and **strictly faithful** if it satisfies both conditions.*

Existing metrics make these notions practical by restricting the set of changes they consider and by implementing less strict measures of change in model output (Sanchez-Lengeling et al., 2020). Specifically, *unfaithfulness* (Unf) (Agarwal et al., 2023) estimates sufficiency as the Kullback-Leibler divergence between the original label distribution and that obtained after zeroing-out all irrelevant features from $G_A$. *Fidelity minus* (Fid-) (Pope et al., 2019; Yuan et al., 2022; Amara et al., 2022) instead erases all edges and features deemed irrelevant by the explanation and measures the change in likelihood of the prediction $\hat{y}$.[4] *Robust fidelity minus* (RFid-) does the same but repeatedly deletes random edges from $C_A$ (Zhao et al., 2023; Amara et al., 2023; Zheng et al., 2023). Finally, *probability of sufficiency* (PS) estimates how often the model's prediction changes after multiplying the node features with the relevance scores output by the detector (Tan et al., 2022). Metrics for necessity are specular, *i.e.*, they manipulate the input by removing *relevant* elements instead, and include *fidelity plus* (Fid+), *robust fidelity plus* (RFid+), and *probability of necessity* (PN).

While existing metrics differ in many details, they nicely fit the same common format:

**Definition 2.** *Let $d$ be a divergence, $p_R$ a distribution over supergraphs of $R_A$, and $p_C$ a distribution over supergraphs of $C_A$. Also, let $\Delta_d(G_A, G'_A) = d(p_\theta(\cdot \mid G_A) \parallel p_\theta(\cdot \mid G'_A))$ measure the impact of replacing $G_A$ with $G'_A$ on the label distribution.[5] Then, the **degree of sufficiency** and **degree of necessity** of an explanation $R_A$ for a decision $(G_A, \hat{y})$ are:*

$$\mathsf{Suf}_{d,p_R}(R_A) = \mathbb{E}_{G'_A \sim p_R}[\Delta_d(G_A, G'_A)], \quad \mathsf{Nec}_{d,p_C}(R_A) = \mathbb{E}_{G'_A \sim p_C}[\Delta_d(G_A, G'_A)] \quad (2)$$

---

[3]We focus on graph classification, but our results also hold for node classification unless otherwise specified.
[4]A variant based on the difference in accuracy also exists (Yuan et al., 2022).
[5]We drop the dependency on the label(s) for brevity.

Table 1: **Definition 2 recovers existing faithfulness metrics** for appropriate choices of divergence $d$ and interventional distributions $p_R$ and $p_C$.

| Metric | Estimates | Divergence $d$ | Allowed changes |
|---|---|---|---|
| Unf | Suf | $\mathsf{KL}(p_\theta(\cdot \mid G_A), p_\theta(\cdot \mid G'_A))$ | zero out all irrelevant features |
| Fid- | | $\lvert p_\theta(\hat{y} \mid G_A) - p_\theta(\hat{y} \mid G'_A) \rvert$ | zero out all irrelevant features, delete all irrelevant edges |
| RFid- | | " | delete a random subset of irrelevant edges |
| PS | | $\mathbb{1}\{p_\theta(\hat{y} \mid G_A) = p_\theta(\hat{y} \mid G'_A)\}$ | multiply all irrelevant elements by relevance scores |
| Fid+ | Nec | $\lvert p_\theta(\hat{y} \mid G_A) - p_\theta(\hat{y} \mid G'_A) \rvert$ | zero out all relevant features, delete all relevant edges |
| RFid+ | | " | delete a random subset of relevant edges |
| PN | | $\mathbb{1}\{p_\theta(\hat{y} \mid G_A) \neq p_\theta(\hat{y} \mid G'_A)\}$ | multiply all relevant elements by relevance scores |

The expectations in Eq. (2) are potentially unbounded, but can be *normalized* to $[0, 1]$, the higher the better, via a non-linear transformation, *i.e.*, taking $\exp(-\mathsf{Suf}_{d,p_R}(R_A))$ and $1 - \exp(-\mathsf{Nec}_{d,p_C}(R_A))$. Table 1 shows that Definition 2 recovers all existing metrics for appropriate choices of divergence $d$ and distributions $p_R$ and $p_C$. Alternative choices yield additional metrics that differ in how they estimate the model's response to input modifications (Appendix C.4).

While *both sufficiency and necessity matter, there exists a natural tension between them.* For instance, explanations covering a larger portion of $G_A$ likely attain higher sufficiency but lower necessity. This motivates us to define the ***degree of faithfulness*** $\mathsf{Faith}_{d,p_R,p_C}(R_A)$ as the harmonic mean of normalized sufficiency and necessity, which is biased towards the lower of the two. We henceforth suppress the dependency on $d$, $p_R$ and $p_C$ whenever it is clear from context.

### 3.1 FAITHFULNESS METRICS ARE NOT INTERCHANGABLE

A first key observation is that, despite falling into a common template, existing metrics are *not interchangeable*, in the sense that explanations that are highly faithful according to one metric can be arbitrarily unfaithful for the others. We formalize this in the following proposition:

**Proposition 1.** *Let $(p_R, p_C)$ be a pair of distributions as per Definition 2. Then, depending on $p_\theta$ and $G_A$, it is possible to find $(p'_R, p'_C)$ such that $\lvert \mathsf{Suf}_{d,p_R}(R_A) - \mathsf{Suf}_{d,p'_R}(R_A) \rvert$ and $\lvert \mathsf{Nec}_{d,p_C}(R_A) - \mathsf{Nec}_{d,p'_C}(R_A) \rvert$ are as large as the natural range of $d$.*

All proofs and relevant discussion are in Appendix A and Appendix C.5. In essence, Proposition 1 means that *faithfulness results cannot be properly interpreted unless the parameters $d$, $p_R$ and $p_C$ are known.* This entails that explanation producers – such as banks and algorithm designers – responsible for certifying the faithfulness of explanations cannot withhold the parameters they used in the computation, lest end-users blindly trust explanations that are not sufficiently faithful for their downstream applications.

To understand the practical implications of our result, we report in Table 2 the $\mathsf{Suf}$ values and ranking of explanations produced by three popular modular GNNs (see Table 4) on the `Motif2` (Gui et al., 2023) dataset. This comes with in-distribution (ID) and out-of-distribution (OOD) splits, allowing us to sample perturbations from different graph distributions. We consider different distributions $p_R$, as follows:

- $p_R^{id_1}$ and $p_R^{ood_1}$ allow *i)* replacing the complement $C_A = G_A \setminus R_A$ of the input graph with that of another sample $G'_A = C'_A \cup R'_A$ taken from the same split, and *ii)* removing random edges from $C_A$.

- $p_R^{id_2}$ and $p_R^{ood_2}$ only subsample the complement of each graph by randomly removing a fixed budget of edges.

Metrics are reported for both ID or OOD splits. Overall, *changing $p_R$ alters the ranking of the models,* This

Table 2: Model ranking and absolute $\mathsf{Suf}$ values for different distributions $p_R$, averaged over 5 seeds: **both can significantly change**.

| Split | Model | Motif2 | |
|---|---|---|---|
| | | $p_R^{id_1}$ | $p_R^{id_2}$ |
| ID | LECI | 1 ($81 \pm 03$) | 2 ($82 \pm 03$) |
| | GSAT | 2 ($78 \pm 01$) | 1 ($84 \pm 02$) |
| | CIGA | 3 ($65 \pm 07$) | 3 ($73 \pm 06$) |
| | | $p_R^{ood_1}$ | $p_R^{ood_2}$ |
| OOD | LECI | 2 ($83 \pm 06$) | 1 ($88 \pm 06$) |
| | GSAT | 3 ($76 \pm 02$) | 3 ($79 \pm 03$) |
| | CIGA | 1 ($85 \pm 09$) | 2 ($86 \pm 03$) |

confirms that care needs to be taken when picking a metric for comparing XAI algorithms and explanations, as metrics are not interchangeable. See Appendix C.5 for additional results.

## 3.2 NOT ALL NECESSITY ESTIMATORS ARE EQUALLY RELIABLE

The non-interchangeability of metrics becomes particularly significant as not all estimators are equally reliable. In fact, as we show next, commonly used necessity metrics are ***insensitive to the number of irrelevant elements in the explanation***. Recall that evaluating necessity involves assessing the model's sensitivity to changes to the explanation $R_A$ itself. Assuming that $R_A$ contains $r$ truly *relevant* edges (*i.e.*, removing them impacts the label distribution) we would like $R_A$'s necessity to worsen as the number of truly *irrelevant* edges $\|R_A\| - r$ it covers grows.

**Necessity metrics are systematically invariant to truly irrelevant edges.** Existing metrics do not satisfy this desideratum. This is because they delete either *all* of $R_A$[6] (*e.g.*, Fid+ and PN) or an IID subset of edges thereof (RFid+). To see why this is problematic, consider a node classification task and a target node $u_1$, and split the graph $G_A$ into two subsets: those nodes whose messages *can* impact the distribution of $Y_1$, and those that *cannot*. For injective $L$-layer GNNs, these sets are $N_L(u_1)$ and $G \setminus N_L(u_1)$, respectively, as there are not enough layers for messages coming from the latter to reach $u_1$.

Now, consider a 1-layer GNN for node classification and an input line graph $u_1 \leftarrow u_2 \leftarrow \cdots \leftarrow u_n$. Since there is only one layer, only the messages from $N_1(u_1) = \{u_1, u_2\}$ can contribute to the distribution of $Y_1$. By deleting all of $R_A$, Fid+ and PN *disconnect $u_1$ from all other nodes*, and as such they cannot distinguish between a "perfect" explanation $R = N_1(u_1)$ and a "bad" explanation $R' = N_n(u_1)$ that contains arbitrary many irrelevant edges, as in both cases the prediction is made using $u_1$ only. See Fig. 2 in the Appendix for an illustration. An analogous reasoning applies to RFid+, which removes edges in an IID fashion. This means that the probability that it deletes the edge between $u_1$ and $u_2$ is the same regardless of how many other edges appear in the explanation. We formalize this observation in the following proposition:

**Proposition 2.** *Fix a divergence $d$ and a threshold $\epsilon > 0$. Let $R$ contain $r$ truly relevant edges. Then, Fid+$(R)$, PN$(R)$, and RFid+$(R)$ do not depend on $\|R\| - r$.*

**How can we ensure that** Nec **depends on the number of irrelevant elements?** We next show that by selecting an appropriate distribution $p_C$, necessity can be made to account for the number of irrelevant edges. This can be achieved by erasing $b$ edges from the explanation, with some constraints on how b is chosen. Specifically, if b depends on $\|R\|$, the metric – once again – does not properly account for irrelevant items in the explanation, as the number of deletions increases according to the size of the explanation. Indeed, a numerical simulation shows that the probability of deleting at least one truly relevant edge does not depend on the number of irrelevant ones in the explanation, if b is proportional to the explanation size (see Fig. 6 in the Appendix). On the other hand, if b is proportional to the size of the input graph $G$, the number of deletions depends also on the size of the complement, resulting in a metric confounded by the complement of the explanation[7]. This is best seen in the following example:

**Example 1.** *Fix an explanation $R$ composed of two edges $\{e_1, e_2\}$, where only $e_1$ is truly relevant. If $G$ has $10$ edges, fixing a budget of deletion as $10\%$ of the graph size yields a modified graph $G'$ such that the probability of removing $e_1$ is $\frac{1}{2}$. Instead, if $G$ has $20$ edges, the probability of removing it is $1$. Hence, despite $R$ being fixed, the probability of sampling truly relevant edges drastically changes.*

We propose to avoid these issues by fixing a budget b that is proportional to a data set-wide statistic, such as average the graph size $\bar{m} = 1/|\mathcal{D}| \sum_{G \in \mathcal{D}} \|G\|$, where $\mathcal{D}$ is the set of graphs. This way, it no longer directly depends on the size of any specific input graph or explanation, while still being adaptive to the target task. The considerations above can be formalized in the following result:

**Proposition 3.** *Fix any divergence $d$ and a constant budget $b \geq 1$. Let $\mathcal{S}_R^b$ be the set of subgraphs of $G$ obtained by deleting $b$ edges from $R$ while keeping $C$ fixed. Given an explanation $R$ containing $r$ truly relevant edges, $\mathsf{Nec}(R)$ computed using a uniform $p_C$ over $\mathcal{S}_R^b$ depends on the number of irrelevant edges $\|R\| - r$.*

---

[6]In node classification, the node whose prediction is being explained is never removed.

[7]The dependency of b on $G$ can be modeled as a causal graph Nec $\leftarrow R_A \leftarrow G \rightarrow b \rightarrow$ Nec, which contains the *backdoor path* $R_A \leftarrow G \rightarrow b \rightarrow$ Nec confounding the estimation of Nec (Pearl, 2009).

Table 4: **Popular modular GNNs fail to fully implement {HS, ER, CF, LA}. ✗/✓** means that both variants exist and the choice is made via cross-validation.

| SE-GNNs | HS | ER | CF | LA | DI-GNNs | HS | ER | CF | LA |
|---|---|---|---|---|---|---|---|---|---|
| GISST (Lin et al., 2020) | ✗ | ✗ | ✓ | ✓ | CIGA (Chen et al., 2022) | TopK | ✓ | ✗/✓ | ✗/✓ |
| GSAT (Miao et al., 2022a) | ✗ | ✗ | ✓ | ✗/✓ | GSAT (Miao et al., 2022a) | ✗ | ✗ | ✓ | ✗/✓ |
| RAGE (Kosan et al., 2023) | ✗ | ✗ | ✓ | ✓ | LECI (Gui et al., 2023) | ✗ | ✗ | ✓ | ✗/✓ |

In essence, *we propose measuring necessity using $p_C$ that is uniform over subgraphs of size $\|G\| - b$*, where b is a hyper-parameter and set as above. We denote the resulting distribution $p_C^b$.

**Experimental analysis**. We empirically compare our proposed Nec metric and RFid+ for assessing the necessity of explanations produced by GSAT (Miao et al., 2022a) trained on Motif2 (Gui et al., 2022). We measure them at two different topK selection thresholds (Amara et al., 2022), measuring how metrics behave for larger explanations. Confirming our theoretical analysis, RFid+ assigns close to constant scores to unnecessarily large explanations (Proposition 2), whereas Nec equipped with our proposed $p_C^b$ sensibly penalizes larger explanations (Proposition 3). Metrics are normalized ensuring the

Table 3: Necessity values for GSAT explanations on Motif2. Nec with $p_C^b$ penalizes larger explanations, whereas RFid+ does not.

| Metric | Top-30% | Top-90% |
|---|---|---|
| RFid+ (↑) | $18 \pm 02$ | $21 \pm 02$ |
| Nec (↑) | $54 \pm 04$ | $34 \pm 02$ |

higher the better, $d$ is set to be the $L_1$ divergence, and values are averaged over 5 seeds. For RFid+, the edge-deletion probability $\kappa = 0.3$, while for Nec the budget $b = 0.3\bar{m}$. We provide more details about metric implementation in Appendix B.2.2 and further comparisons in Appendix D.2.

# 4 Is Faithfulness Worth Optimizing For?

We show that *for regular injective GNNs, strict faithful explanations are trivial*, highlighting a trade-off between explainability and expressivity. For modular GNNs, instead, non-trivial strictly faithful explanations are theoretically attainable provided certain conditions are met, see Appendix C.2. We then verify that popular modular GNNs fail in fulfilling those conditions, highlighting strong limitation in their current design principles.

## 4.1 For injective regular GNNs strictly faithful explanations are trivial

Recall that an explanation $R_A$ is *strictly faithful* when no change to the complement $C_A$ affects the model's output, and when all elements of $R_A$ contribute to the final prediction (see Section 3). In the following, we show that for regular GNNs strictly faithful explanations are *completely uninformative*. To build intuition, notice that strictly sufficient explanations must subsume the computational graph of the prediction, that is, the subgraph of $G_A$ induced by all nodes whose messages influence the label distribution. In node classification, this is the $L$-hop neighborhood of the target node, and in graph classification the whole input graph. More formally:

**Proposition 4.** *Consider a binary classification task, an $L$-layer injective GNN, any $p_C$ and $p_R$ not allowing the addition of new elements, and d being either a divergence between distributions or the difference in prediction likelihood: i) for **node classification** that only uses local aggregators, an explanation $R_A$ for a decision $(G_A, \hat{y}_u)$ is strictly faithful iff it matches $N_L(u)$. ii) For **graph classification**, an explanation $R_A$ for a decision $(G_A, \hat{y})$ is strictly faithful iff it matches $G_A$.*

In both cases *strictly faithful explanations do not depend on the learned weights at all* and, as such, are uninformative. This is a direct consequence of injectivity, a prerequisite for GNNs to implement the Weisfeiler-Lehman test (Xu et al., 2018; Bianchi and Lachi, 2024), and highlights a trade-off between expressivity and explainability.[8] This finding also complements existing negative results on (even non-injective) regular GNNs, which provide no easy way of obtaining strictly faithful explanations (Amara et al., 2022; Longa et al., 2024; Li et al., 2024).

---

[8]Of course, one can obtain more informative explanations by giving up on strict faithfulness and instead aiming at producing explanations that are both faithful enough and smaller than the entire computational graph.

Table 5: Test set accuracy and faithfulness of SE-GNNs augmented with the strategies outlined in Section 4.2, averaged over 5 seeds. We do not augment `GISST` on `BaMS`, `Motif2`, and `Motif-Size` since these datasets have constant input features for all nodes, making the modifications ineffective.

| Dataset | BaMS | | Motif2 | | Motif-Size | | BBBP | |
|---|---|---|---|---|---|---|---|---|
| | Acc | Faith | Acc | Faith | Acc | Faith | Acc | Faith |
| `GSAT` | $100 \pm 00$ | $35 \pm 03$ | $92 \pm 01$ | $61 \pm 01$ | $90 \pm 01$ | $60 \pm 02$ | $79 \pm 04$ | $27 \pm 08$ |
| `GSAT` + **ER** | $100 \pm 00$ | $35 \pm 03$ | $92 \pm 01$ | $63 \pm 01$ | $90 \pm 01$ | $65 \pm 01$ | $80 \pm 02$ | $33 \pm 04$ |
| `GSAT` + **HS** | $98 \pm 01$ | $21 \pm 06$ | $53 \pm 02$ | $24 \pm 05$ | $54 \pm 03$ | $22 \pm 05$ | $71 \pm 01$ | $31 \pm 09$ |
| `GSAT` + **ER** + **HS** | $99 \pm 01$ | $24 \pm 04$ | $57 \pm 04$ | $37 \pm 03$ | $56 \pm 07$ | $29 \pm 09$ | $73 \pm 02$ | $33 \pm 02$ |
| `GISST` | $100 \pm 00$ | $25 \pm 03$ | $92 \pm 01$ | $53 \pm 02$ | $92 \pm 00$ | $50 \pm 02$ | $84 \pm 03$ | $23 \pm 11$ |
| `GISST` + **ER** | – | – | – | – | – | – | $85 \pm 06$ | $27 \pm 06$ |
| `GISST` + **HS** | – | – | – | – | – | – | $83 \pm 05$ | $19 \pm 07$ |
| `GISST` + **ER** + **HS** | – | – | – | – | – | – | $81 \pm 07$ | $15 \pm 09$ |
| `RAGE` | $96 \pm 01$ | $33 \pm 05$ | $83 \pm 02$ | $64 \pm 04$ | $74 \pm 09$ | $63 \pm 07$ | $82 \pm 01$ | $33 \pm 04$ |
| `RAGE` + **ER** | $96 \pm 02$ | $33 \pm 02$ | $85 \pm 06$ | $66 \pm 03$ | $71 \pm 09$ | $55 \pm 07$ | $84 \pm 01$ | $33 \pm 05$ |
| `RAGE` + **HS** | $97 \pm 01$ | $46 \pm 03$ | $85 \pm 01$ | $65 \pm 02$ | $78 \pm 07$ | $65 \pm 09$ | $84 \pm 02$ | $46 \pm 02$ |
| `RAGE` + **ER** + **HS** | $96 \pm 01$ | $46 \pm 04$ | $83 \pm 04$ | $64 \pm 04$ | $75 \pm 08$ | $62 \pm 12$ | $82 \pm 01$ | $43 \pm 03$ |

## 4.2 THE CASE OF MODULAR GNNS

Modular GNNs are explicitly designed so that their predictions – at least on paper – depend solely on the explanation they extract, yet popular modular architectures suffer from poor faithfulness (Christiansen et al., 2023). To better understand this phenomenon, we analyze four "architectural desiderata", whose absence enables information to *leak from the complement to the explanation*, allowing the complement to influence the label distribution and thus compromising faithfulness:

- **Content Features** (**CF**): since the message passing of the detector $f$ is unconditional on its predicted edge relevance score, the classifier $g$ should have access to raw input (or content) features, contrarily to using the detector's latent representation which is heavily leaked;

- **Hard Scores** (**HS**): the detector $f$ should associate exact zero importance to information in the complement, thus preventing update and aggr from mixing information from $R_A$ and $C_A$ in $g$;

- **Explanation Readout** (**ER**): (for graph classification only) the final graph global readout should aggregate only over $R_A$. Since explanations are often soft edge masks over the entire graph, scaling the node embedding according to their average importance scores can still reduce the impact of irrelevant nodes;

- **Local Aggregations** (**LA**): non-local aggregations (see Section 2) can easily mix the information of the explanation with that of its complement, and can create unwanted dependencies between any pair of nodes in the graph.

All these architectural desiderata encourage or enforce the classifier $g$ to rely solely on $R_A$ for making predictions. In the remainder of this Section, we empirically investigate the impact on faithfulness of integrating popular models with the desiderata they lack, reporting their accuracy as a sanity check.

**Experimental analysis**. We benchmarked six representative modular architectures, listed in Table 4, following their respective evaluation testbed for graph classification. The datasets chosen for evaluation are picked from the usual evaluation routines for SE-GNNs and DI-GNNs respectively. For DI-GNNs, the chosen benchmarks can be divided into datasets with known domain-invariant input motifs (`Motif2-Basis`, `CMNIST-Color`), datasets with presumed domain-invariant input motifs (`LBAPcore`), and datasets where domain-invariant input motif are not expected (`SST2`). This last case is especially problematic, as no clear advantage is expected in focusing on a subset of the input graph. The full experimental setup is detailed in Appendix B.

Table 5 and Table 6 show the results for the original SE-GNNs and DI-GNNs architectures and their augmented variants. Overall, we show that simple but often neglected architectural design choices can greatly impact faithfulness. Our major findings are as follows:

**(i) Local explanations are not always helpful**. `LBAPcore` and `SST2` are the hardest datasets to improve on and, generally, have worse faithfulness and accuracy scores. This raises doubts about the

Table 6: OOD test set accuracy and faithfulness of DI-GNNs augmented with the strategies delineated in Section 4.2, averaged over 5 seeds.

| Dataset | Motif2 | | CMNIST | | LBAPcore | | SST2 | |
|---|---|---|---|---|---|---|---|---|
| | Acc | Faith | Acc | Faith | Acc | Faith | Acc | Faith |
| LECI | $85 \pm 07$ | $58 \pm 02$ | $26 \pm 10$ | $48 \pm 11$ | $71 \pm 01$ | $43 \pm 05$ | $83 \pm 01$ | $26 \pm 02$ |
| LECI + **ER** | $86 \pm 03$ | $59 \pm 02$ | $58 \pm 12$ | $58 \pm 03$ | $71 \pm 01$ | $46 \pm 02$ | $82 \pm 01$ | $13 \pm 02$ |
| LECI + **HS** | $86 \pm 04$ | $57 \pm 02$ | $34 \pm 10$ | $57 \pm 01$ | $72 \pm 01$ | $24 \pm 01$ | $83 \pm 01$ | $18 \pm 03$ |
| LECI + **LA** | - | - | $46 \pm 11$ | $57 \pm 03$ | $69 \pm 01$ | $31 \pm 03$ | $81 \pm 05$ | $20 \pm 04$ |
| LECI + **ER + HS + LA** | $79 \pm 11$ | $55 \pm 02$ | $75 \pm 06$ | $61 \pm 01$ | $59 \pm 02$ | $21 \pm 01$ | $81 \pm 02$ | $16 \pm 01$ |
| CIGA | $46 \pm 10$ | $38 \pm 08$ | $23 \pm 03$ | $36 \pm 03$ | $69 \pm 01$ | $33 \pm 02$ | $76 \pm 06$ | $18 \pm 01$ |
| CIGA + **ER** | $45 \pm 09$ | $54 \pm 04$ | $23 \pm 02$ | $43 \pm 07$ | $59 \pm 07$ | $09 \pm 06$ | $74 \pm 03$ | $16 \pm 01$ |
| CIGA + **CF** | $53 \pm 07$ | $49 \pm 02$ | $13 \pm 01$ | $49 \pm 09$ | $49 \pm 12$ | $03 \pm 01$ | $55 \pm 07$ | $09 \pm 12$ |
| CIGA + **LA** | - | - | $30 \pm 09$ | $41 \pm 03$ | $68 \pm 01$ | $34 \pm 08$ | $79 \pm 03$ | $16 \pm 01$ |
| CIGA + **ER + CF + LA** | $47 \pm 08$ | $39 \pm 07$ | $23 \pm 03$ | $50 \pm 02$ | $66 \pm 01$ | $23 \pm 10$ | $76 \pm 08$ | $15 \pm 02$ |
| GSAT | $75 \pm 06$ | $58 \pm 01$ | $25 \pm 04$ | $48 \pm 03$ | $70 \pm 03$ | $40 \pm 06$ | $79 \pm 04$ | $22 \pm 04$ |
| GSAT + **ER** | $59 \pm 06$ | $60 \pm 06$ | $30 \pm 06$ | $48 \pm 01$ | $67 \pm 01$ | $32 \pm 03$ | $81 \pm 01$ | $23 \pm 04$ |
| GSAT + **HS** | $86 \pm 03$ | $42 \pm 07$ | $14 \pm 03$ | $44 \pm 04$ | $71 \pm 01$ | $28 \pm 01$ | $80 \pm 02$ | $17 \pm 02$ |
| GSAT + **LA** | - | - | $27 \pm 03$ | $46 \pm 04$ | $69 \pm 03$ | $44 \pm 02$ | $81 \pm 01$ | $21 \pm 04$ |
| GSAT + **ER + HS + LA** | $64 \pm 08$ | $47 \pm 03$ | $17 \pm 02$ | $51 \pm 05$ | $70 \pm 01$ | $38 \pm 03$ | $80 \pm 01$ | $17 \pm 01$ |

suitability of these – rather popular – datasets for evaluating local GNN explanations, as evidenced by the fact that explanations tend to cover the entire graph with similar or nearly identical scores (see Appendix D.1 for the details). Indeed, a standard GIN (Xu et al., 2018) trained without any OOD regularization achieves scores 69.7 and 79.7 respectively, confirming that GNNs might not have an advantage in focusing on a sparse input subgraph. Therefore, any strategy that pushes the model to more closely align with the explanation is likely to force it to either discard information or assign uniform weights across the entire topology, ultimately reducing both faithfulness and accuracy.

**(ii) HS can significantly alter model behavior**. Although RAGE seems to benefit from **HS**, as it improves both faithfulness and accuracy in every dataset, it severely compromises the training of GSAT in Motif2 and Motif-Size, resulting in a train and test accuracy of around $50\%$. In fact, exact-zero scores for irrelevant edges are certainly desirable, yet it is known that it is hard to deal with exact zeros in common gradient-based learning. More advanced techniques are therefore advocated for reliably implementing hard scores (Serra and Niepert, 2022).

**(iii) ER, LA, and CF can be effective in promoting faithfulness**. Motivated by these observations, next we focus on the remaining datasets, where GNNs *can* identify sensible local explanations. **ER** improves faithfulness in 10 cases out of 15, while leaving it unchanged in four. In two cases in particular, whilst faithfulness increases, accuracy drops. This is a natural effect of being faithful: if the explanation is not plausible enough[9], the accuracy shall drop. When prompting CIGA to use input features (**CF**), faithfulness increases by $33\%$ on Motif2 and CMNIST on average. Similarly, **LA** improves on average by $9.5\%$ faithfulness and by $38\%$ accuracy on CMNIST. In general, imposing all desiderata jointly is beneficial when individually they also are.

**Take-away**. On the one hand, these results *question the current design of modular GNNs*, suggesting that popular models include architectural components that implicitly penalize faithfulness, and that the factors contributing to faithfulness may be more nuanced than previously assumed.

On the other, they show that merely preventing information from the complement to leak into the prediction is insufficient to fully address the problem. A rather complementary direction of investigation is that of making the detector $f$ more stable to irrelevant modifications. In fact, assuming a classifier perfectly implementing the desiderata above and an explanation $R_A$, it is easy to see that if the detector changes the predicted explanation for a modified graph $G'_A \supseteq R_A$, then the final prediction can arbitrarily change, thus also Suf does.

Training-time strategies are also likely to play a role. For example, necessity can intuitively be encouraged via sparsification techniques (Lin et al., 2020; Kosan et al., 2023), whereas approaches like contrastive learning (Chen et al., 2022; 2023), adversarial training (Gui et al., 2023), or data

---

[9]CIGA and GSAT score around 0.25 of WIoU plausibility on Motif2. See Appendix B.2 for details

augmentation (Wu et al., 2022) are natural options for encouraging sufficiency. We plan to investigate how those insights translate into new faithful-by-design modular architectures in future work.

## 5 FAITHFULNESS IS KEY FOR DOMAIN INVARIANCE

Finally, we study the impact of faithfulness – and lack thereof – on domain invariance. In the domain invariance literature (Chen et al., 2022; Gui et al., 2023), one distinguishes between in-distribution (ID) and out-of-distribution (OOD) data, sampled from $p^{id}(G_A, Y)$ and $p^{ood}(G_A, Y)$, respectively, and the goal is to learn a DI-GNN $p_\theta$ that generalizes from ID to OOD data. Input graphs are assumed to consist of an invariant $R_A^* \subseteq G_A$ and a spurious subgraph, with the ground-truth label only depending on the invariant subgraph in a domain-invariant manner, that is, $p^{id}(Y \mid R_A^*) = p^{ood}(Y \mid R_A^*)$ (Cai et al., 2023; Chen et al., 2023; Jiang et al., 2023). Modular DI-GNNs are designed to output a subgraph $R_A$ (which plays the role of an explanation) and its degree of invariance is proportional to its ***plausibility***, *i.e.*, how closely it matches $R_A^*$.

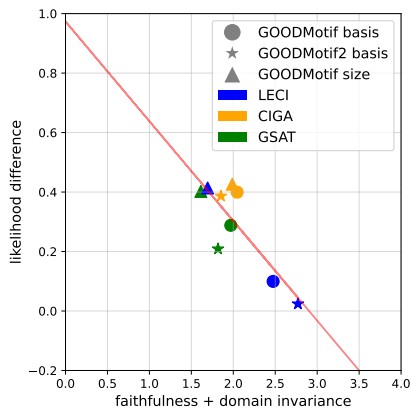

Figure 1: **Likelihood, faithfulness and domain-invariance are correlated**. The plot shows the difference in likelihood between splits. The red line is the best linear fit. Best viewed in color.

We begin by showing that, even if the graphs $R_A$ extracted by a DI-GNN (see Section 2) are maximally plausible, unless they are also strictly sufficient, the model's prediction is *not* domain invariant. To build intuition, take a DI-GNN that, for some input, outputs a perfectly invariant explanation $R_A$. Now, if the explanation is not strictly sufficient, by definition there exists a modification to the complement of $R_A$ (which is domain-dependent) that alters the predicted class distribution. Thus the model as a whole cannot be domain-invariant.

**Proposition 5.** *Let $p_\theta$ be a modular DI-GNN such that the detector $f$ outputs graphs $R_A$ that are maximally plausible, i.e., that comprise all and only those elements that are constant across the target domains. If $R_A$ is not strictly sufficient, then the prediction is not domain invariant.*

This result is significant because DI-GNNs typically optimize only for the invariance of the extracted subgraph $R_A$, neglecting how the classifier uses it afterward. The next result highlights that the degree of faithfulness of $R_A$ – and specifically, that of sufficiency – plays a direct role in ensuring that the model's predictions are truly domain-invariant. In fact, the following Theorem bounds the difference in likelihood between ID and OOD as a proxy to measure the change in the fit of the data, so that a small difference corresponds to a consistent model behavior across domains. Differently from the many generalisation bounds available in literature (Redko et al., 2020), ours pertains specifically to explanation quality, both in terms of degree of invariance and degree of sufficiency.

**Theorem 1.** *(Informal) Let $p_\theta$ be a deterministic DI-GNN with detector $f$ and classifier $g$, and $p^{id}(G_A, Y)$ and $p^{ood}(G_A, Y)$ be the ID and OOD empirical distributions, respectively. Then:*

$$\left| \mathbb{E}_{(G_A,y)\sim p^{id}}[p_\theta(y \mid G_A)] - \mathbb{E}_{(G_A,y)\sim p^{ood}}[p_\theta(y \mid G_A)] \right| \tag{3}$$
$$\leq \mathbb{E}_{R_A^*} \left[ k_1(\lambda_{topo}^{id} + \lambda_{topo}^{ood}) + k_2(\lambda_{feat}^{id} + \lambda_{feat}^{ood}) + (\lambda_{suff}^{id} + \lambda_{suff}^{ood}) \right].$$

*Here, $\mathbb{E}_{R_A^*}$ runs over all possible invariant subgraphs, $k_1, k_2 > 0$ are Lipschitz constants of $p_\theta$, $\lambda_{topo}^{id}$, $\lambda_{topo}^{ood}$ $\lambda_{feat}^{id}$, and $\lambda_{feat}^{ood}$ measure the implausibility of the detector's output (with respect to topology and features, respectively), and $\lambda_{suff}^{id}, \lambda_{suff}^{ood}$ are the degree of sufficiency for ID and OOD data.*

The Theorem is proved by applying the triangular inequality and basic properties of the expectation and relies on two main assumptions: the domain invariance of $R_A^*$, and the Lipschitzness of GNNs. In essence, Theorem 1 shows that a DI-GNN that fits well the ID data will fit well the OOD data if $R_A$ is i) plausible (low $\lambda_{topo}$ and $\lambda_{feat}$, thus invariant across domains), and ii) highly sufficient (low

$\lambda_{suff}$). An illustration of the interplay between the LHS and the RHS of Theorem 1 is provided in Appendix C.1. The practical relevance of this result is twofold: it suggests that researchers should also consider sufficiency when evaluating DI-GNNs, as well as focus on the invariance of both topology and feature, rather than topology only as often done (see Table 8).

We stress that while necessity does not appear in the bound, it becomes important if we wish the invariant subgraph to be non-trivial. Indeed, Fig. 1 shows a statistically significant anti-correlation (Pearson's correlation $-0.83$, $p$-value $0.01$) between the difference in average prediction's likelihood between ID and OOD data, and the combination of the degree of domain-invariance and faithfulness. Notably, the same applies, yet with a slightly weaker correlation of $-0.74$ (p-value $0.02$), when replacing likelihood with accuracy. See Appendix B for further details.

## 6  RELATED WORK

**Modular architectures.** Modular GNNs are widely used for seeking trustworthiness, either in the form of explainable (Lin et al., 2020; Miao et al., 2022b; Kosan et al., 2023) or confounding-free predictions (Wu et al., 2022; Chen et al., 2022; Gui et al., 2023) (see Jiang et al. (2023) for a survey), also in deep neural networks for non-relational data (Koh et al., 2020; Marconato et al., 2022). Explanation quality is critical for interpretability and for ensuring fair and generalizable predictions (Amara et al., 2022; Miao et al., 2022a; Longa et al., 2024). Despite being primarily designed for faithfulness, as the prediction depends on the explanation only, little attention has been paid to measuring *how well* the classifier adheres to its own explanations, an issue that can completely prevent trust allocation (Agarwal et al., 2024). Indeed, recent work has shown how for popular SE-GNNs architectures, the classifier seems to be more faithful to randomly generated subgraphs than to its explanations (Christiansen et al., 2023). In this work, we investigate the reasons for this behavior, discuss the applicability of faithfulness-enforcing strategies and extend the analysis to DI-GNNs.

**Faithfulness.** In the Explainable AI (Samek et al., 2021) and GNN literature (Agarwal et al., 2023), sufficiency and faithfulness are often conflated, however, disregarding necessity opens to trivially sufficient yet uninformative explanations. Faithfulness is also sometimes defined as the correlation between the explainer's "performance" and that of the model, that is, to what extent worsening model performance worsen also the plausibility of explanations (Sanchez-Lengeling et al., 2020). However, this is evaluating model and explainer performance separately and does not reflect the degree to which a classifier relies on the provided explanation for making its prediction. Our notion of faithfulness is rooted in causal explainability (Beckers, 2022) and disentanglement in causal representation learning (Suter et al., 2019; Schölkopf et al., 2021). Like Beckers (2022), we argue that absolute measurements of sufficiency, and more generally faithfulness, are uninformative unless properly contextualized, in our case by the choice of parameters used for the measurements. Following Bordt et al. (2022), we argue this leaves room for adversarial explanation providers to supply explanations that optimize their objective rather than that of the explanation consumer.

## 7  KEY TAKEAWAYS AND BROADER IMPACT

We have studied the faithfulness of explanations in regular and modular GNNs. Our results indicate that, despite conforming to a shared template, ***existing faithfulness metrics are not interchangeable*** (Proposition 1) and in fact some even suffer from systematic issues (Proposition 2), which we show can be avoided with an appropriate choice of parameters (Proposition 3). We have also shown that ***optimizing for faithfulness is not always a sensible design goal*** (Proposition 4) and that improving the faithfulness of SE-GNNs and DI-GNNs is non-trivial, suggesting limitations in the current design of modular GNN. We also proved that faithfulness plays a surprisingly ***central role for domain invariance***, in that modular GNNs tailored for domain invariance cannot be invariant unless their explanations are also sufficient (Proposition 5 and Theorem 1). This suggests that research on DI-GNNs should focus on sufficiency, which it currently neglects.

**Limitations:** Our experiments explore simple yet overlooked design choices to enhance faithfulness in modular GNNs, showing benefits for SOTA architectures. Developing truly faithful-by-design solutions remains a challenge for future work. In addition, our notion of faithfulness relies on input-level modifications of the graph topology. While this doesn't impact training or inference time, its evaluation can be slow for large input graphs, requiring more efficient implementations.

ETHICS STATEMENT

All authors have read and approved the ICLR Code of Ethics. As for societal consequences, the aim of this work is to shed light on the non-trivial nature of explanation faithfulness, warning against its unqualified reporting and interpretation. It can thus contribute to the development of truly interpretable and trustworthy models for networked data.

REPRODUCIBILITY STATEMENT

The proofs of all theoretical results are provided in Appendix A. Details about the experiments, along with details about metrics and implementation, are available in Appendix B. Our source code is made available in the supplemental.

ACKNOWLEDGEMENTS

Funded by the European Union. Views and opinions expressed are however those of the author(s) only and do not necessarily reflect those of the European Union or the European Health and Digital Executive Agency (HaDEA). Neither the European Union nor the granting authority can be held responsible for them. Grant Agreement no. 101120763 - TANGO.

The authors are grateful to Emanuele Marconato for useful discussions and to Manfred Jaeger for his feedback on an earlier version of this manuscript.

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

## A    PROOFS

### A.1    PROOF OF PROPOSITION 1

**Proposition 1.** *Let $(p_R, p_C)$ be a pair of distributions as per [Definition 2](#). Then, depending on $p_\theta$ and $G_A$, it is possible to find $(p'_R, p'_C)$ such that $|\mathsf{Suf}_{d,p_R}(R_A) - \mathsf{Suf}_{d,p'_R}(R_A)|$ and $|\mathsf{Nec}_{d,p_C}(R_A) - \mathsf{Nec}_{d,p'_C}(R_A)|$ are as large as the natural range of $d$.*

*Proof.* We prove the statement for graph classification; the case of node classification is analogous.

Let $\mathcal{S} = \mathrm{supp}(p_R)$ and $\mathcal{S}' = \mathrm{supp}(p'_R)$, both finite. Assume $R_A$ attains a perfect degree of sufficiency according to $p_R$, *i.e.*, $\mathsf{Suf}_{p_R}(R_A) = 0$, and take $p'_R$ such that $\mathcal{S} \cap \mathcal{S}' = \varnothing$. Then:

$$|\mathsf{Suf}_{p_R}(R_A) - \mathsf{Suf}_{p'_R}(R_A)| = \Big| \sum_{G'_A \in \mathcal{S}} \Delta(G_A, G'_A) p_R(G'_A) + \sum_{G'_A \in \mathcal{S}'} \Delta(G_A, G'_A) p'_R(G'_A) \Big| \quad (4)$$

The two sums have no terms in common, so we can independently control the second one through our choice of interventional distribution $p'_R$. In fact, it is maximized by those distributions that associate all probability mass to graphs $G'_A$ for which $\Delta(G_A, G'_A)$ is largest, *e.g.*, counterfactuals. As long as at least one counterfactual $G'_A$ exists that includes $R_A$ as a subgraph, then $p'_R$ can allocate all probability mass to it, in which case the right sum becomes as large as $\mathrm{argmax}_{\mathbf{p},\mathbf{q}} d(\mathbf{p}, \mathbf{q})$, that is, as large as the natural range of the divergence $d$.

A similar reasoning applies to Nec. Take $R_A$ such that for all $R'_A \in \mathrm{supp}(p_C)$ it holds that $\Delta(G_A, G'_A) > \tau$ for some desired threshold $\tau$. As long as there exists a graph $G'_A \supseteq C_A$ such that $p_\theta(\cdot \mid G_A) \equiv p_\theta(\cdot \mid G'_A)$, then we can always choose an interventional distribution $p'_C$ that allocates all mass to it. Hence, $\mathsf{Nec}_{p'_C}(R_A) = 0$ and the difference in necessity will be at least $\tau$.    $\square$

This simple result warrants some discussion. First, we note that the assumptions it hinges on are rather weak. Its two key requirements are that: i) $\mathcal{S}$ and $\mathcal{S}'$ are finite: this is the case for all existing metrics, whose interventional distributions are defined over the set of *subgraphs* of the input graph, which are finitely many; ii) The GNN admits counterfactuals that subsume $R_A$ (for sufficiency) or inputs that subsume $C_A$ and map to the predicted label $\hat{y}$ (for necessity), both frequent occurrences in practice.

Second, the construction behind the proof also mimics actual differences between faithfulness metrics from the literature. In fact, ignoring differences in choice of divergence, the main difference between unfaithfulness (which allocates non-zero probability only to subgraphs obtained by deleting features from the input), fidelity minus (which deletes all edges), and robust fidelity minus (which deletes a random subset of edges at random) is exactly their interventional distributions $p_R$, which also happen to have either disjoint or almost disjoint supports (in which case a similar result applies almost verbatim). Our construction then amounts to saying that there are many practical situations in which one can achieve good unfaithfulness and poor fidelity minus, or vice versa.

Let us briefly discuss the relationship between metrics having the same interventional distribution but different divergences $d$ and $d'$. On the bright side, it is easy to see that, as long as both $d$ and $d'$ are proper divergences, an explanation $R_A$ achieving perfect sufficiency according to one will also attain perfect sufficiency according to the other, precisely because perfect sufficiency is attained when $\Delta(G_A, G'_A) = 0$ for all $G'_A \sim p_R$, which can occur if and only if both divergenes are zero (by definition of divergence). The same holds for the degree of necessity. For non-optimal sufficiency and necessity, the difference due to replacing divergences is governed by well-known inequalities, such as Hölder's (between $L_p$ distances) and Pinsker's (relating the KL to the $L_1$ distance).

The situation is different if $d$ is a proper divergence (say, the KL divergence) and $d'$ is a difference in likelihoods. In this case, there are situations in which the two quantities can differ. To see this, consider a multi-class classification problem, target decision $(G_A, \hat{y})$ and a modified input $G'_A \sim p_R$ such that i) $p_\theta(\hat{y} \mid G_A) = p_\theta(\hat{y} \mid G'_A)$, but ii) the two label distributions have disjoint support. In this case, the KL divergence between them would be unbounded, yet the difference in likelihood would be null. Depending on the choice of $p_R$, this can yield a large difference between degrees of sufficiency (and similarly for necessity).

## A.2 PROOF OF PROPOSITION 2

**Proposition 2.** *Fix a divergence $d$ and a threshold $\epsilon > 0$. Let $R$ contain $r$ truly relevant edges. Then, $\mathsf{Fid+}(R)$, $\mathsf{PN}(R)$, and $\mathsf{RFid+}(R)$ do not depend on $\|R\| - r$.*

*Proof.* $\mathsf{Fid+}$ **and** $\mathsf{PN}$ **are systematically invariant to truly irrelevant edges.** We begin by showing that metrics like $\mathsf{Fid+}$, which estimate necessity by deleting the whole explanation $R_A$,[10] suffer from this issue. Consider a $L$-layer GNN for predicting a label of interest for a node $u$, and take $R_A \subseteq N_L(u)$ which covers all truly relevant edges and $R'_A \supset R_A$. $\mathsf{Fid+}$ associates the same value to both $R_A$ and $R'_A$, as in both cases the model predicts $Y_1$ using $u$ only, meaning that, surprisingly, $\mathsf{Fid+}(R_A) = \mathsf{Fid+}(R'_A)$. When the edge relevance scores are binary, $\mathsf{PN}$ behaves exactly like $\mathsf{Fid+}$, in which case it also suffers from this issue.

$\mathsf{RFid+}$ **is systematically invariant to truly irrelevant edges.** Considering the same setting as above, it turns out that also $\mathsf{RFid+}(R_A) = \mathsf{RFid+}(R'_A)$. In fact, $\mathsf{RFid+}$ estimates necessity by removing edges in an IID fashion, which means that the probability that it deletes the edges in $R_A$ is the same regardless of how many other edges appear in $R'_A \setminus R_A$. This intuition can be formalized as follows: Let $U_j$ be a random variable determining whether edge $e_j$ is kept, such that $U_j \sim Bernoulli(\kappa)$, with $\kappa$ a user-provided hyperparameter. As in Proposition 3, let $\mathcal{S}_R^b$ be the set of subgraphs of $G$ obtained by deleting $b$ edges from $R$ while keeping $C$ fixed, and $\mathcal{A}_R^b = \{G' \in \mathcal{S}_R^b : \Delta(G, G') \geq \epsilon\}$ those subgraphs that lead to a large enough change in $\Delta$. Also, let $\mathcal{S}_R = \mathcal{S}_R^1 \cup \ldots \cup \mathcal{S}_R^m$ and similarly for $\mathcal{A}_R$. Then, we can study $\mathsf{RFid+}$ in terms of how likely the corrupted graphs $G'$ it samples have missing relevant edges, *i.e.*, $P(G' \in \mathcal{A}_R)$.

$$P(G' \in \mathcal{A}_R) = 1 - P(G' \notin \mathcal{A}_R) \tag{5}$$
$$= 1 - P(U_1 = 1, \ldots, U_m = 1) \tag{6}$$
$$= 1 - \prod_{j=1}^{m} P(U_j = 1) \tag{7}$$
$$= 1 - \kappa^m \tag{8}$$
$$\tag{9}$$

In the second to last step, we made use of the independence of all $U_i$'s. The above probability does not depend on the number of irrelevant edges $\|R\| - r$. $\qquad\square$

## A.3 PROOF OF PROPOSITION 3

**Proposition 3.** *Fix any divergence $d$ and a constant budget $b \geq 1$. Let $\mathcal{S}_R^b$ be the set of subgraphs of $G$ obtained by deleting $b$ edges from $R$ while keeping $C$ fixed. Given an explanation $R$ containing $r$ truly relevant edges, $\mathsf{Nec}(R)$ computed using a uniform $p_C$ over $\mathcal{S}_R^b$ depends on the number of irrelevant edges $\|R\| - r$.*

*Proof.* Let $\mathcal{S}_R^b$ be the set of subgraphs of $G$ obtained by deleting $b$ edges from $R$ while keeping $C$ fixed, and $\mathcal{A}_R^b = \{G' \in \mathcal{S}_R^b : \Delta(G, G') \geq \epsilon\}$ those subgraphs that lead to a large enough change in $\Delta$. Then, when choosing a budget $b$ of deletions and a uniform $p_C(G)$ over subgraphs with $\|G\| - b$ edges, $P(G' \in \mathcal{A}_R^b) = |\mathcal{A}_R^b|/|\mathcal{S}_R^b|$, which decreases as the number of irrelevant edges in $R$ increases. Note that $|\mathcal{S}_R^b| = \binom{\|R\|}{b}$, while $|\mathcal{A}_R^b| = \sum_{c=1}^{b} \binom{r}{c}\binom{\|R\|-r}{r-c}$. $\qquad\square$

## A.4 PROOF OF PROPOSITION 4

**Proposition 4.** *Consider a binary classification task, an $L$-layer injective GNN, any $p_C$ and $p_R$ not allowing the addition of new elements, and $d$ being either a divergence between distributions or the difference in prediction likelihood: i) for **node classification** that only uses local aggregators, an explanation $R_A$ for a decision $(G_A, \hat{y}_u)$ is strictly faithful iff it matches $N_L(u)$. ii) For **graph classification**, an explanation $R_A$ for a decision $(G_A, \hat{y})$ is strictly faithful iff it matches $G_A$.*

---

[10]In node classification, the node whose prediction is being explained is never removed.

*Proof.* Before proceeding, it is useful to introduce the notion of computational graph, that is, the subgraph of $G_A$ induced by those nodes whose messages are relevant for determining the label distribution $P(Y_u \mid G_A)$ or $P(Y \mid G_A)$. For node classification, it is easy to see that for GNNs with local aggregators, the only messages reaching $u$ are those coming from nodes within $N_L(u)$, and for injective GNNs, all such messages impact the label distribution. For graph classification, which is implicitly global, all nodes in the computational graph are the entirety of $G_A$.

Next, consider any pair of interventional distribution, $p_C$ and $p_C$, excluding the addition of new nodes or edges. We show that if $R_A$ does not subsume the entirety of the computational graph, then when computing strict sufficiency we can always alter those elements of the computation graph that fall in the complement $C_A$. By injectivity of $p_\theta$, any change to the features or edges of the computational graph yields a difference in label distribution, necessarily affecting both classes and therefore any divergence or difference in likelihood $d$[11]. It follows that $R_A$ cannot be strictly sufficient for any $d$, and therefore neither strictly faithful.

The converse also holds: if $R_A$ covers the entire computational graph, then it is necessarily strictly sufficient. This is because the complement $C_A$ has no overlap with the computational graph, so altering it has no impact on the label distribution.

If instead, the interventional distributions allow for the addition of new items, like new nodes or new edges, then a strictly faithful explanation might not exist altogether as the induced perturbation can adversarially include the trivial class-discriminative pattern of the non-predicted class, forcing the model to always switch prediction. □

### A.5 PROOF OF PROPOSITION 5

**Proposition 5.** *Let $p_\theta$ be a modular DI-GNN such that the detector $f$ outputs graphs $R_A$ that are maximally plausible, i.e., that comprise all and only those elements that are constant across the target domains. If $R_A$ is not strictly sufficient, then the prediction is not domain invariant.*

*Proof.* We proceed by contradiction. Let $G_A$ be an input graph such that the explanation $R_A$ for model $p_\theta$ and decision $(G_A, \hat{y})$ is perfectly domain-invariant yet not strictly sufficient. We assume that the model prediction does not depend on domain-induced spurious information, meaning that every modification applied to $C_A$ will not have any impact on the model prediction (according to $d$). However, the lack of strict sufficiency implies that $\exists G'_A \in \text{supp}(p_R)$ such that $\Delta(G_A, G_A') > 0$. This means that there exists a perturbation outside of $R_A$ that altered the model prediction, which is in contradiction with assuming that it does not depend on domain-induced information. □

### A.6 PROOF OF THEOREM 1

**Theorem:** *Let $p_\theta$ be a deterministic DI-GNN with detector $f$ and classifier $g$ and $d$ be the difference in likelihood of the predicted label. Also, let $p_\theta$ be Lipshitz with respect to changes to both the topology and the features of the input, that is, for every pair of graphs $G_A = (G, X)$ and $G'_A = (G', X')$, it must hold that:*

$$\left| p_\theta(Y \mid (G, X)) - p_\theta(Y \mid (G', X)) \right| \leq k_1 d_1(G, G') \tag{10}$$

$$\left| p_\theta(Y \mid (G, X)) - p_\theta(Y \mid (G, X')) \right| \leq k_2 d_2(X, X') \tag{11}$$

*for suitable distance functions $d_1$ and $d_2$. Then:*

$$\left| \mathbb{E}_{(G_A, y) \sim p^{id}}[p_\theta(y \mid G_A)] - \mathbb{E}_{(G_A, y) \sim p^{ood}}[p_\theta(y \mid G_A)] \right| \tag{12}$$

$$\leq \mathbb{E}_{R_A^*}[k_1(\lambda_{topo}^{id} + \lambda_{topo}^{ood}) + k_2(\lambda_{feat}^{id} + \lambda_{feat}^{ood}) + (\lambda_{suff}^{id} + \lambda_{suff}^{ood})]$$

*Here, $k_1, k_2 > 0$ are Lipschitz constants of $p_\theta$, $\lambda_{topo}^{id}$, $\lambda_{topo}^{id}$ $\lambda_{feat}^{ood}$, and $\lambda_{feat}^{ood}$ measure the (negated) degree of domain invariance of the detector's output (with respect to topology and features, respectively), and $\lambda_{suff}^{id}, \lambda_{suff}^{ood}$ are the degree of sufficiency for ID and OOD data.*

---

[11]It is not affecting the difference in accuracy however, as a change in prediction confidence does not always entail a change in most-likely output class.

*Proof.* Let $f^*$ be an *ideal* detector that for every input $G_A$ outputs the maximal domain-invariant subgraph $R_A^* = (R^*, X^{R^*})$. Note that, for every $(G_A^{id}, y^{id}) \sim p^{id}$ and $(G_A^{ood}, y^{ood}) \sim p^{ood}$ that share the same invariant subgraph (that is, such that $f^*(G_A^{id}) = f^*(G_A^{ood}) = R_A^*$) any DI-GNN constructed by stitching together $f^*$ and a deterministic classifier $g$ will predict the same label distribution for both graphs, *i.e.*, $p_g(Y \mid f^*(G_A^{id})) \equiv p_g(Y \mid f^*(G_A^{ood}))$.

Now, fix any invariant subgraph $R_A^*$. When conditioning on it, it holds:

$$\mathbb{E}_{(G_A, y) \sim p^{id}(G_A, Y \mid R_A^*)} [p_\theta(y \mid f^*(G_A^{id}))] = \mathbb{E}_{(G_A, y) \sim p^{ood}(G_A, Y \mid R_A^*)} [p_\theta(y \mid f^*(G_A^{ood}))] \quad (13)$$

where the two expectations run over ID and OOD samples that share the same invariant subgraph $R_A^*$. We proceed by noting that:

$$\mathbb{E}_{(G_A, y) \sim p^{id}(\cdot \mid R_A^*)} \left| p_\theta(y \mid (R^*, X^{R^*})) - p_\theta(y \mid (R, X^{R^*})) \right| \leq k_1 \underbrace{\mathbb{E}_{(G_A, y) \sim p^{id}(\cdot \mid R_A^*)} [d_1(R^*, R)]}_{:= \lambda_{\text{topo}}} \quad (14)$$

The expectation on the RHS is the average topological distance of the predicted invariant subgraphs to $R_A^*$, which corresponds to the degree of invariance (plausibility) of $R$, and will be referred to as $\lambda_{\text{topo}}$. At the same time, it also holds that:

$$\mathbb{E}_{(G_A, y) \sim p^{id}(\cdot \mid R_A^*)} \left| p_\theta(y \mid (R, X^{R^*})) - p_\theta(y \mid (R, X^R)) \right| \leq k_2 \underbrace{\mathbb{E}_{(G_A, y) \sim p^{id}(\cdot \mid R_A^*)} [d_2(X^{R^*}, X^R)]}_{:= \lambda_{\text{feat}}} \quad (15)$$

where the expectation on the RHS is now the average distance between the features of the predicted invariant subgraphs and those of $R_A^*$, and will be referred to as $\lambda_{\text{feat}}$. Combining Eq. (14) and Eq. (15) using the triangular inequality, yields:

$$\mathbb{E}_{(G_A, y) \sim p^{id}(\cdot \mid R_A^*)} \left| p_\theta(y \mid R_A^*) - p_\theta(y \mid R_A) \right| \quad (16)$$

$$= \mathbb{E}_{(G_A, y) \sim p^{id}(\cdot \mid R_A^*)} \left| p_\theta(y \mid R_A^*) - p_\theta(y \mid (R, X^{R^*})) + p_\theta(y \mid (R, X^{R^*})) - p_\theta(y \mid R_A) \right| \quad (17)$$

$$\leq \mathbb{E} \left| p_\theta(y \mid (R^*, X^{R^*})) - p_\theta(y \mid (R, X^{R^*})) \right| + \mathbb{E} \left| p_\theta(y \mid (R, X^{R^*})) - p_\theta(y \mid (R, X^R)) \right| \quad (18)$$

$$\leq k_1 \lambda_{\text{topo}}^{id} + k_2 \lambda_{\text{feat}}^{id} \quad (19)$$

This holds for any fixed $R_A^*$.

Next, we bound the expected difference between the label distribution determined by $R_A$ and that determined by $G_A$ using the degree of sufficiency. For notational convenience, we draw complements $C_A'$ rather than full modified graphs $G_A'$ from $p_R$, and we denote the operation of joining $R_A$ and $C_A'$ to form $G_A'$ with the $\cup$ operator, and where we use the difference in prediction likelihood as divergence $d$. We proceed as follows:

$$\mathbb{E}_{(G_A, y) \sim p^{id}(\cdot \mid R_A^*)} \left| p_\theta(y \mid R_A) - p_\theta(y \mid G_A) \right| \quad (20)$$

$$= \mathbb{E}_{(G_A, y) \sim p^{id}(\cdot \mid R_A^*)} \left| \mathbb{E}_{C_A' \sim p_R(G_A)} [p_\theta(y \mid R_A \cup C_A')] - p_\theta(y \mid G_A) \right| \quad (21)$$

$$\leq \mathbb{E}_{(G_A, y) \sim p^{id}(\cdot \mid R_A^*)} \mathbb{E}_{C_A' \sim p_R(G_A)} \left| p_\theta(y \mid R_A \cup C_A') - p_\theta(y \mid G_A) \right| \quad (22)$$

$$= \mathbb{E}_{(G_A, y) \sim p^{id}(\cdot \mid R_A^*)} \text{Suf}(R_A) := \lambda_{\text{suff}} \quad (23)$$

$$\quad (24)$$

In the first to second step, we made use of the law of total probability and the product rule, for which:

$$p_\theta(y \mid R_A) = \sum_{C_A'} p(y, C_A' \mid R_A) = \sum_{C_A'} \frac{p(y, C_A', R_A)}{p(R_A)} = \sum_{C_A'} \frac{p(y \mid C_A', R_A)p(C_A' \mid R_A)p(R_A)}{p(R_A)} \tag{25}$$

$$= \sum_{C_A'} p_\theta(y \mid R_A \cup C_A')p_R(C_A') = \mathbb{E}_{C_A' \sim p_R} p_\theta(y \mid R_A \cup C_A') \tag{26}$$

Then, applying again the triangular inequality between Eq. (20) and Eq. (16) yields:

$$\mathbb{E}_{(G_A, y) \sim p^{id}(\cdot \mid R_A^*)} \left| p_\theta(y \mid R_A^*) - p_\theta(y \mid G_A) \right| \leq k_1 \lambda_{\text{topo}}^{id} + k_2 \lambda_{\text{feat}}^{id} + \lambda_{\text{suff}}^{id}. \tag{27}$$

The same derivation applies to $p^{ood}(G_A)$, so we obtain:

$$\mathbb{E}_{(G_A, y) \sim p^{ood}(\cdot \mid R_A^*)} \left| p_\theta(y \mid R_A^*) - p_\theta(y \mid G_A) \right| \leq k_1 \lambda_{\text{topo}}^{ood} + k_2 \lambda_{\text{feat}}^{ood} + \lambda_{\text{suff}}^{ood}, \tag{28}$$

Combining these two bounds using the triangular inequality one last time, we derive:

$$\left| \mathbb{E}_{(G_A, y) \sim p^{id}(\cdot \mid R_A^*)} p_\theta(y \mid G_A) - \mathbb{E}_{(G_A, y) \sim p^{ood}(\cdot \mid R_A^*)} p_\theta(y \mid G_A) \right| \tag{29}$$

$$\leq \mathbb{E}_{\substack{(G_A^{id}, y) \sim p^{id}(\cdot \mid R_A^*) \\ (G_A^{ood}, y) \sim p^{ood}(\cdot \mid R_A^*)}} \left| p_\theta(y \mid G_A^{id}) - p_\theta(y \mid G_A^{ood}) \right| \tag{30}$$

$$\leq k_1(\lambda_{\text{topo}}^{id} + \lambda_{\text{topo}}^{ood}) + k_2(\lambda_{\text{feat}}^{id} + \lambda_{\text{feat}}^{ood}) + (\lambda_{\text{suff}}^{id} + \lambda_{\text{suff}}^{ood}). \tag{31}$$

Again, this holds for all choices of $R_A^*$.

Finally, we leverage these inequalities to bound the difference in likelihood between ID and OOD data for all possible choices of $R_A^*$:

$$\mathbb{E}_{R_A^*} \left[ \left| \mathbb{E}_{(G_A, y) \sim p^{id}(\cdot \mid R_A^*)} p_\theta(y \mid G_A) - \mathbb{E}_{(G_A, y) \sim p^{ood}(\cdot \mid R_A^*)} p_\theta(y \mid G_A) \right| \right] \tag{32}$$

$$\geq \left| \mathbb{E}_{R_A^*} \left[ \mathbb{E}_{(G_A, y) \sim p^{id}(\cdot \mid R_A^*)} p_\theta(y \mid G_A) - \mathbb{E}_{(G_A, y) \sim p^{ood}(\cdot \mid R_A^*)} p_\theta(y \mid G_A) \right] \right| \tag{33}$$

$$= \left| \mathbb{E}_{(G_A, y) \sim p^{id}} p_\theta(y \mid G_A) - \mathbb{E}_{(G_A, y) \sim p^{ood}} p_\theta(y \mid G_A) \right| \tag{34}$$

where $\mathbb{E}_{R_A^*}$ runs over all possible invariant subgraphs and no longer depends on the specific choice of $R_A^*$. By monotonicity of the expectation we conclude that:

$$\left| \mathbb{E}_{(G_A, y) \sim p^{id}} p_\theta(y \mid G_A) - \mathbb{E}_{(G_A, y) \sim p^{ood}} p_\theta(y \mid G_A) \right| \tag{35}$$

$$\leq \mathbb{E}_{R_A^*} \left[ k_1(\lambda_{\text{topo}}^{id} + \lambda_{\text{topo}}^{ood}) + k_2(\lambda_{\text{feat}}^{id} + \lambda_{\text{feat}}^{ood}) + (\lambda_{\text{suff}}^{id} + \lambda_{\text{suff}}^{ood}) \right] \tag{36}$$

$$\square$$

# B  EXPERIMENTAL DETAILS

## B.1  DATASETS

In this study, we conduct an investigation across seven graph classification datasets commonly used for evaluating SE-GNNs and DI-GNNs. Specifically, we examine three synthetic datasets and four real-world datasets, and in the following paragraphs, we provide a detailed description of each.

**Synthetic datasets**

- `BaMS` (Azzolin et al., 2022) is a synthetic dataset consisting of 1,000 Barabasi-Albert (BA) graphs, each with network motifs (house, grid, wheel) randomly attached at various positions. Class 0 includes plain BA graphs and BA graphs enriched with either a house, a grid, a wheel, or all three motifs together. Class 1 consists of BA graphs enriched with a combination of two motifs: a house and a grid, a house and a wheel, or a wheel and a grid. This dataset is utilized in the context of self-explainable Graph Neural Networks (GNNs), with the expected ground truth explanations being the specific motif combinations in each class.

- `Motif2-Basis` (Gui et al., 2023) is a synthetic dataset comprising 24,000 graphs categorized into three classes. Each graph consists of a basis and a motif. The basis can be a ladder, a tree (or a path), or a wheel. The motifs are a house (class 0), a five-node cycle (class 1), or a crane (class 2). The dataset is divided into training (18,000 graphs), validation (3,000 graphs), and test sets (3,000 graphs). In the context of OOD analysis, two additional sets are considered: the OOD validation set and the OOD test set. In these sets, the class-discriminative subgraph $R_A$ (i.e., the motifs) remains fixed, while the bases vary. Specifically, the basis for the OOD validation set (3,000 graphs) is a circular ladder, and the basis for the OOD test(3,000 graphs) set is a Dorogovtsev-Mendes graph (Dorogovtsev and Mendes, 2002).

- `Motif-Basis` (Gui et al., 2022) is specular to `Motif2-Basis`, where the OOD test set contains graphs generated connecting the three motifs above to simple line paths of varying length.

- `Motif-Size` (Gui et al., 2022) is a synthetic dataset consisting of 24,000 graphs categorized into three classes. Similar to `Motif2-Basis`, each network is composed of a basis and a motif, with the motif serving as the class-discriminative subgraph $R_A$. The basis structures include a ladder, a tree (or a path), a wheel, a circular ladder, or a star. The motifs are a house (class 0), a five-node cycle (class 1), or a crane (class 2). The dataset is divided into training (18,000 graphs), validation (3,000 graphs), and test sets (3,000 graphs). In the context of OOD analysis, the size of the basis is increased. Specifically, in the OOD validation set (3,000 networks), the basis sizes are increased up to three times their original size, while in the OOD test set (3,000 graphs), the basis sizes are increased up to seven times.

**Real-world datasets**

- `BBBP` (Wu et al., 2018) is a dataset derived from a study on modeling and predicting barrier permeability (Martins et al., 2012). It comprises 2,050 compounds, with 483 labeled as positive and 1,567 as negative.

- `CMNIST-Color` (Gui et al., 2022) contains 70,000 graphs of hand-written digits transformed from the MNIST database using superpixel techniques (Monti et al., 2017). The digits are colored according to their domains and concepts. In the training set, which contains 50,000 graphs, the digits are colored using five different colors. To evaluate the model's performance on out-of-distribution data, the validation and testing set each contain 10,000 graphs with two new colors introduced specifically for these sets.

- `LBAPcore-Assay` (Gui et al., 2023) is a molecular dataset consisting of 34,179 graphs sourced from the 311 largest chemical assays. The validation set comprises 19,028 graphs from the next largest 314 assays, while the test set includes 19,302 graphs from the smallest 314 assays.

- `SST2-Length` is a sentiment analysis dataset based on the NLP task of sentiment analysis, adapted from the work of Yuan et al. (Yuan et al., 2022). In this dataset, each sentence is transformed into a grammar tree graph, where individual words serve as nodes and their associated word embeddings act as node features. The primary task is a binary classification to predict the sentiment polarity of each sentence. The dataset comprises 70,042 graphs, divided into training, validation, and test sets. The out-of-distribution (OOD) validation and test sets are specifically created to evaluate performance on data with longer sentence lengths.

## B.2 IMPLEMENTATION DETAILS

### B.2.1 TRAINING AND REPRODUCIBILITY

The models are developed leveraging repositories provided by previous work. Specifically:

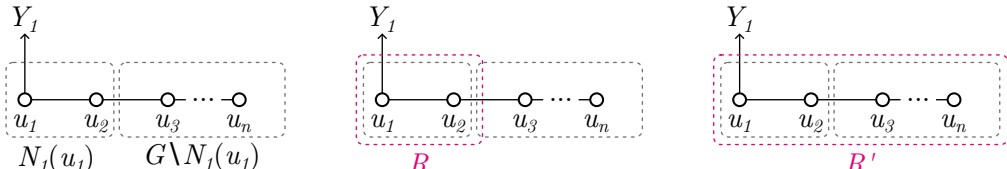

Figure 2: **Existing necessity metrics are invariant to the number of truly irrelevant edges** that they cover. For a line graph and a 1-layer GNN, the *truly* relevant edges for predicting the label of $u_1$ are those in $N_1(u_1) = \{u_1, u_2\}$. However, existing necessity metrics do not distinguish between $R$ (which contains no truly irrelevant edge) and a dummy $R'$ (which contains arbitrarily many).

- CIGA, LECI and GSAT are developed based on the repository from Gui et al. (2023), using commit fb39550453b4160527f0dcf11da63de43a276ad5.

- RAGE is implemented using the code available at https://anonymous.4open.science/r/RAGE as described in Kosan et al. (2023).

- For GISST, we use the repository from https://anonymous.4open.science/r/SEGNNEval, following the approach outlined in Lin et al. (2020).

To encourage reproducibility, we stick to the hyperparameters provided in each respective repository, except for GSAT on BBBP and BaMS where we set the values of ood_param to 0.5 and extra_param to [True, 10, 0.2], and for Motif2-Basis and Motif-Size in Table 5 where we set the values of ood_param to 10 and extra_param to [True, 10, 0.2]. To favor a fair comparison with the other architectures, we changed the RAGE's default GNN backbone to GIN (Xu et al., 2018), using a *mean* global readout for the final prediction. Model selection was performed on the ID validation set.

### B.2.2 COMPUTING FAITHFULNESS

- **Necessity**: Nec follows the guidelines presented in Section 3.2. The idea is that choosing a fixed budget depending, for example, on a dataset-wide statistic is a simple yet sensible choice for a necessity metric properly accounting for the number of irrelevant edges present in an explanation. Specifically, we chose the budget b as a fixed proportion of the average number of undirected edges for each split of the dataset, where the proportion ratio is set to $5\%$ in all our experiments. For each explanation, we sample a number $q_1$ of perturbed graphs, where $q_1$ is fixed at $8$ in our experiments and each modification deletes a random edge.

- **Sufficiency**: For Suf, instead, we apply two types of perturbations: *(i)* replace the complement of the explanation with that of a random sample from the same data split, joining with random connections the original explanation with the sampled complement; *(ii)* delete random edges from the complement, as prescribed by Nec. We repeat these interventions for a number $q_1$ of samples for each explanation, where $q_1$ is fixed at $8$ in our experiments, as for Nec. Since the joining operation is performed explicitly at the input level, the resulting procedure can become computationally heavy, especially for large graphs. Alternative approaches try to emulate input interventions with latent interventions, where manipulations to the embedding of explanation and complement are applied, respectively (Wu et al., 2022; Sui et al., 2022). We argue that this type of latent intervention is suboptimal for a correct evaluation of faithfulness, as the latent vectors used for interventions are not guaranteed to sufficiently disentangle explanation and complement information, resulting in leakages.

- **Faithfulness**: Faith corresponds to the harmonic mean of the normalized Nec and normalized Suf scores, where we set $d$ as the $L_1$ divergence for both metrics, and it is computed over a subset of $q_2$ of input graphs, which is set to $800$ in our experiments. Since most modular GNNs output soft edge scores, we extract the relevant subgraph via TopK selection, where the size ratios vary in $\{0.3, 0.6, 0.9\}$. Then, the resulting Faith is taken as the best value across ratios. An ablation study to compare the resulting scores across a varying number of samples $q_2$ and interventions $q_1$ is provided in Fig. 10 and Fig. 11. In all experiments, we use the normalized Suf and Nec, such that the values are the higher the better.

**Are faithfulness metrics well calibrated?** In general, all metrics outlined in Table 1 are not calibrated, in the sense that a value of, e.g., $0.5$ does not have the same meaning across metrics. This issue is naturally present also in Suf and Nec, and can impair metric comparison across different hyper-parameters. Nonetheless, this is not affecting our empirical investigation as we always stick to the same hyper-parameters.

### B.2.3 COMPUTING THE DEGREE OF INVARIANCE (PLAUSIBILITY)

The degree of invariance appearing in Theorem 1 represents how close the predicted subgraph $R_A$ is to the truly domain-invariant input motif, which is assumed to be the only stable factor enabling for stable predictions across domains. Carrying over the terminology from XAI, we compute this quantity as the *plausibility* (Longa et al., 2024) of the provided subgraph with respect to a known ground truth. Specifically, we compute the topology-wise degree of invariance as the Weighted Intersection over Union (WIoU) between the provided explanation relevance score, and the expected ground truth scores, which equal $1$ for the invariant edges in the graph, and $0$ for the others. The feature-wise degree of invariance, instead, is unmeasured as the ground truth over invariant features is typically overlooked, and no settled evaluation testbed is available. The same is done for computing the plausibility of SE-GNNs with respect to a known ground truth explanation.

### B.2.4 IMPLEMENTATION OF FAITHFULNESS-ENFORCING STRATEGIES

As stated in Section 4.2, we included the aforementioned architectural desiderata in several popular models to verify their impact on model accuracy and faithfulness. In the following, we describe how the four desiderata are implemented in practice:

- **Hard Scores** (**HS**) To enforce the generation of binary 0-1 explanation masks by the detector $f$, we apply a technique similar to the Straight-Through (ST) trick used in the discrete Gumbel-Softmax Estimator (Jang et al., 2017). Specifically, during the forward pass, we utilize the binary version of the mask, while in the backward pass, we use the continuous version.

- **Content Features** (**CF**) To ensure the classifier uses only raw input features, we simply replace the feature matrix with that of the input data before feeding it to the classifier $g$.

- **Explanation Readout** (**ER**) The Explanation Readout strategy computes the graph embedding by applying a global aggregator encouraging the classifier to adhere more to $R_A$. This involves multiplying the node mask $M$, obtained by averaging over incident nodes the predicted edge mask, with the node embedding before performing the final global readout, i.e., $\mathbf{h}_G = \mathrm{aggr}_G(\{\{M_u \mathbf{h}_u^L : u \in V\}\})$. Since explanations are soft scores over $G_A$, this approach ensures the model adheres more closely to the soft mask.

- **Local** `aggr` (**LA**) To enhance the expressivity of Graph Neural Networks (GNNs), some models incorporate virtual nodes (Barceló et al., 2020; Hu et al., 2020; Sestak et al., 2024). We remove those to mitigate the risk of mixing the information of the explanation with that of its complement, which can create unwanted dependencies between pairs of nodes in the graph.

### B.2.5 CHANGES WITH RESPECT TO THE ORIGINAL CODEBASE

Here we describe two minor changes we did to the original codebase.

- **Stable TopK & Permutation-Invariant Metrics.** We found the original implementation of the topK operator to exhibit instabilities when used on GPU, in particular in the presence of equal scores for which alternatively the first or the last elements of the tensor are returned. This results in order-dependent metric values, either over -or under-estimated according to the order of ground truth edges in the graph[12].

  To avoid this bias and to have more predictable behavior, we switched to a stable implementation of the topK operator which always selects elements with the same score as they appear in the input tensor, and we randomly permute nodes and edges in each graph at loading time.

---

[12]In synthetic datasets, ground truth edges are typically the last elements as they are attached to an already generated base-graph.

- **Undirected Explanation Scores.** Accordingly to the original version of the codebase intro-duced in Training and Reproducibility paragraph, `LECI`, and `GSAT` are averaging the edge attention scores among directions for each undirected edge via `torch_sparse.transpose`. However, we found that, even by sticking to the original package versioning, the way `torch_sparse.transpose` is used is not weighting edge scores as expected, but rather is acting as an identity mapping. To fix this bug and to produce undirected edge scores without changing the edge attention mechanism, we average the edge scores via `torch_geometric.to_undirected`. More details are available in our codebase.

  Table 7 compares performance scores for directed vs undirected explanation scores showing comparable performance aggregated over all datasets and models.

Table 7: Model performance scores for directed vs undirected explanation scores. Models labeled with '(D)' use directed edge scores.

| Dataset / Model | LECI | CIGA | GSAT | LECI (D) | CIGA (D) | GSAT (D) |
|---|---|---|---|---|---|---|
| Motif-Basis | $72 \pm 06$ | $42 \pm 02$ | $57 \pm 03$ | $82 \pm 05$ | $50 \pm 05$ | $52 \pm 04$ |
| Motif2-Basis | $85 \pm 07$ | $46 \pm 10$ | $75 \pm 06$ | $81 \pm 06$ | $40 \pm 03$ | $77 \pm 05$ |
| Motif-Size | $41 \pm 06$ | $43 \pm 05$ | $51 \pm 03$ | $54 \pm 06$ | $47 \pm 02$ | $52 \pm 04$ |
| SST2 | $83 \pm 01$ | $76 \pm 06$ | $79 \pm 04$ | $83 \pm 01$ | $77 \pm 04$ | $81 \pm 02$ |
| LBAPcore | $71 \pm 00$ | $69 \pm 01$ | $70 \pm 00$ | $72 \pm 00$ | $70 \pm 01$ | $71 \pm 00$ |
| CMNIST | $26 \pm 10$ | $23 \pm 03$ | $25 \pm 04$ | $28 \pm 17$ | $21 \pm 03$ | $38 \pm 04$ |

## C  FURTHER DISCUSSION

### C.1  DOES EXPLANATION INVARIANCE ENTAIL INVARIANT MODELS?

To build intuition, take a DI-GNN that, for some input, outputs a perfectly invariant explanation $R_A$. Now, if the explanation is not strictly sufficient, by definition there exists a modification to the complement of $R_A$ (which is domain-dependent) that alters the predicted class distribution. Thus the model as a whole cannot be domain-invariant.

This naturally fits Theorem 1. If the detector outputs perfectly invariant explanations $\lambda_{topo}$ and $\lambda_{feat}$ at the RHS of Eq. (3) are zero, yet the sufficiency term $\lambda_{suff}$ can still be larger than zero, meaning that the model can fail invariance (LHS > 0, e.g., it might fit ID data properly but OOD data poorly).

We illustrate the relationship between the RHS and LHS in Fig. 3, which shows that despite `CIGA` having more invariant explanations than `GSAT`, it has worse sufficiency and they tend to fit ID and OOD with a comparable shift. The plots report the LHS (difference in ID and OOD fit) and RHS (invariance/plausibility and sufficiency) of nine DI-GNNs used in our experiments on two data sets. The x-axis shows that `CIGA` (yellow) tends to be more domain-invariant (plausible) than `GSAT` (green) despite having worse sufficiency. The y-axis shows that these models tend to fit ID and OOD data very differently, especially `CIGA` despite its explanations being more invariant (plausible) than `GSAT`'s.

As expected, **explanation invariance does not entail models are domain invariant**.

### C.2  CAN WE ACHIEVE STRICT FAITHFULNESS IN MODULAR GNNS?

In Proposition 4 we provide a theoretical argument regarding the impossibility of injective regular GNNs to achieve strict faithfulness. *Can we provide a similar result for modular GNNs as well?* We can answer this question with a constructive argument: Assume a perfectly stable detector $f$, *i.e.*, it always predicts the same $R_A$ for all $G'_A \sim p_R$. Assume also an injective classifier $g$ such that it implements **HS**, **CF**, and **ER**, as described in Section 4.2. Then, intuitively, the classifier uses no information outside of $R_A$, as the features it is aggregating solely belong to the explanation itself. Then, as $R_A$ is not changing after perturbations of the complement by the stability assumption, every deterministic GNN will achieve strict sufficiency. Then, by injectivity of $g$, every modification to $R_A$ will change the model output, meaning $R_A$ is also strictly necessary.

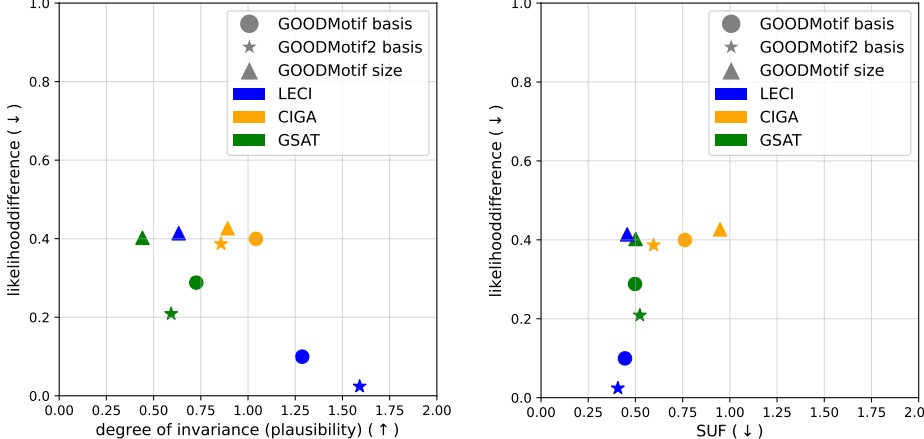

Figure 3: The plots shows the LHS (difference in ID and OOD likelihood) and RHS (invariance/plausibility and sufficiency) of Theorem 1 for nine DI-GNNs used in our experiments on three data sets. The x-axis shows that `CIGA` (orange) tends to be more domain-invariant (plausible) than `GSAT` (green) despite having worse sufficiency. The y-axis shows that these models tend to fit ID and OOD data very differently, especially `CIGA` despite its explanations being more invariant (plausible) than `GSAT` 's. The only model achieving considerably low likelihood difference is `LECI` (blue) for `Motif2-Basis` and `Motif-Basis`, which scores the best both in explanation invariance and Suf. ↓ (↑) stands for the lower (the higher) the better.

This nice result, however, hinders a number of implementation challenges, some of which are already discussed in Section 4.2. In addition, also the possibility of learning truly injective GNN classifiers has recently been questioned (Jaeger, 2023), making this result more of a conceptual blueprint than an actual model design.

Table 8: Many popular DI-GNNs neglect the invariance of node features ($\lambda_{\text{feat}}$).

| Model | Node Feature Invariance |
|---|:---:|
| `DIR` (Wu et al., 2022) | ✗ |
| `VGIB` (Yu et al., 2022) | ✗ |
| `OODGAT` (Song and Wang, 2022) | ✗ |
| `GIL` (Li et al., 2022) | ✗ |
| `GSAT` (Miao et al., 2022a) | ✗ |
| `LRI` (Miao et al., 2022b) | ✓ |
| `CIGA` (Chen et al., 2022) | ✗ |
| `CAL` (Sui et al., 2022) | ✗ |
| `RIGNN` Luo et al. (2023) | ✗ |
| `GALA` (Chen et al., 2023) | ✗ |
| `LECI` (Gui et al., 2023) | ✓ |
| `DIsC` (Fan et al., 2024) | ✗ |
| `GSINA` (Ding et al., 2024) | ✗ |

## C.3 TESTING FOR NON-STRICT SUFFICIENCY

Proposition 5 showed that strict sufficiency is a necessary condition for a DI-GNNs to be domain invariant. In particular, even if an explanation $R_A$ captures the desired invariant subgraph, if it is not strictly sufficient then the prediction will not be domain invariant. Testing for non-strict sufficiency then amounts to finding a counter-example of modification outside of $R_A$ that leads to a change in model output, as measure by the chosen $d$. In Algorithm 1 we illustrate a simple algorithm for testing

non-strict sufficiency, which given as input the input graph $G_A$, the predicted invariant subgraph $R_A$, the chosen distribution $p_R$ specifying the set of allowed modifications, and a budget of perturbations $\sigma$, it returns True if $R_A$ is not strictly sufficient. If the algorithm returns False, then the explanation might still be non strictly sufficient, unless $\sigma \geq |p_R|$ and each element of $p_R$ is sampled at least once. More efficient algorithms are possible, for example by adapting informed perturbation search under constraints as done in adversarial perturbation search (Gosch et al., 2024).

---

**Algorithm 1** Testing for non strict sufficiency

---

**Require:** Input graph $G_A = C_A \cup R_A$, predicted invariant subgraph $R_A$, chosen distribution $p_R$, budget of perturbations $\sigma$, threshold $\tau$
**Ensure:** Returns True if the explanation is non strictly sufficient
1: **for** $i = 1$ to $\sigma$ **do**
2:      Sample $C'_A \sim p_R$
3:      $G'_A = C'_A \cup R_A$
4:      **if** $\Delta_d(G_A, G'_A) \geq \tau$ **then**
5:          **return** True
6:      **end if**
7: **end for**
8: **return** False

---

### C.4 CHOOSING FAITHFULNESS PARAMETERS IS TASK DEPENDENT

In light of Definition 2 and Table 1, selecting a faithfulness metric then amounts to choosing a suitable $p_R$, $p_C$, and $d$. Proposition 1 shows that different parameters are not interchangeable, begging the question of how to properly choose the appropriate metric. In this section, we argue that the choice of metric is application dependent: *stricter* metrics make sense in high-stakes applications, where faithfulness is a must, and for providing adversarial robustness guarantees (for, e.g., Machine Learning as a Service), while *looser* metrics are more meaningful if adversarial inputs are not a concern. There is a clear monotonic relationship between these, in that being strictly sufficient (resp. faithful) to a *stricter* metric implies being strictly sufficient (resp. faithful) to the *looser* one. To see this, consider two different choices for $p_R$:

- **Strictest choice**: $p_R^1$ allows the modification of the complement of the explanation in every conceivable way via node/edge addition and removal and arbitrary feature perturbations. Being strictly sufficient wrt $p_R^1$ means being invariant to any possible change to the complement.

- **Loosest choice**: $p_R^2$ allows only node/edge removal. This is the most narrow distribution, and it corresponds to the typical setting of sufficiency metrics presented in Table 1. $p_R^2$ finds a reasonable use in controlled and low-stakes applications, where only minor perturbations are expected.

While a dual argument holds for $p_C$, we showed in Section 3.2 that some necessity metrics are surprisingly flawed, in that they do not capture the desired semantics. Regarding the choice of $d$ instead, consider a similar distinction:

- **Strictest choice**: $d^1$ is any divergence between class probability distributions, like $L_1$ or KL. It accounts for any change in the model's outputs, and it is thus desirable in scenarios where the ranking of classes matters, for example in conformal prediction sets for Human Decision Making (Cresswell et al.).

- **Intermediate choice**: $d^2$ is the difference in prediction likelihood, and will only account for changes in the confidence of the predicted class. Thus, $d^2$ is more suitable in scenarios in which only the most likely prediction is relevant.

- **Loosest choice**: $d^3$ is the difference in accuracy or most likely prediction. It only accounts for changes in the most likely class, neglecting any change in the model's confidence.

Those insights apply to both post-hoc and modular GNN settings.

## C.5 FAITHFULNESS METRICS ARE NOT INTERCHANGEABLE

In this Section we complement the empirical analysis of Table 2, showing the results also for `Motif-Size` and `CMNIST`.

Table 9: Ranking of models and absolute Suf values according to different interventional distributions $p_R$, averaged over 5 random seeds. Results confirm that **faithfulness measures *can* differ only based on the reference distribution**, as both rankings and absolute values can significantly change.

| # **Ranking** (Suf) | | **Motif-Size** | | **Motif2** | | **CMNIST** | |
|---|---|---|---|---|---|---|---|
| | | $p_R^{id_1}$ | $p_R^{id_2}$ | $p_R^{id_1}$ | $p_R^{id_2}$ | $p_R^{id_1}$ | $p_R^{id_2}$ |
| ID | LECI | 2 (79 ± 06) | 2 (83 ± 06) | 1 (81 ± 03) | 2 (82 ± 03) | 1 (72 ± 19) | 1 (82 ± 18) |
| | GSAT | 1 (83 ± 02) | 1 (86 ± 01) | 2 (78 ± 01) | 1 (84 ± 02) | 2 (56 ± 13) | 2 (59 ± 01) |
| | CIGA | 3 (70 ± 04) | 3 (72 ± 04) | 3 (65 ± 07) | 3 (73 ± 06) | 3 (30 ± 05) | 3 (43 ± 07) |
| | | $p_R^{ood_1}$ | $p_R^{ood_2}$ | $p_R^{ood_1}$ | $p_R^{ood_2}$ | $p_R^{ood_1}$ | $p_R^{ood_2}$ |
| OOD | LECI | 1 (81 ± 06) | 1 (89 ± 05) | 2 (83 ± 06) | 1 (88 ± 06) | 1 (68 ± 19) | 1 (81 ± 13) |
| | GSAT | 2 (73 ± 01) | 2 (80 ± 01) | 3 (76 ± 02) | 3 (79 ± 03) | 2 (50 ± 07) | 2 (54 ± 13) |
| | CIGA | 3 (55 ± 05) | 3 (70 ± 08) | 1 (85 ± 09) | 2 (86 ± 03) | 3 (35 ± 10) | 3 (45 ± 07) |

For `Motif-Size` and `Motif2`, in two cases out of four just changing the type of perturbation allowed by $p_R$ (replacing the complement vs removing random edges; across columns) changes the ranking of models. Instead, changing the graph distribution (ID vs OOD; across rows) alters the ranking in every case. For `CMNIST` instead, the rankings are stable across distributions, even if absolute values can still sensibly change. Overall, these results confirm that different metrics yielded by different distributions are not interchangeable.

## D FURTHER EXPERIMENTS

### D.1 FAILURE CASES

Table 10: Model performance scores for `SST2` and `LBAPcore` on ID validation set and OOD test set.

| Dataset / Model | GIN | | LECI | | GSAT | |
|---|---|---|---|---|---|---|
| | ID val. | OOD test | ID val. | OOD test | ID val. | OOD test |
| `SST2` | 91 ± 00 | 80 ± 01 | 92 ± 00 | 83 ± 01 | 91 ± 00 | 79 ± 04 |
| `LBAPcore` | 92 ± 00 | 70 ± 00 | 92 ± 00 | 71 ± 01 | 92 ± 01 | 70 ± 03 |

As discussed in Section 4.2, `SST2-Length` and `LBAPcore` are two particularly challenging dataset for improving faithfulness. We claim this may come from the task benefiting from global information being used, which clashes with the goal of modular architectures that seek sparse and local discriminative subgraphs. Consider, in fact, the results reported in Table 10 where even a simple *regular* GNN baseline trained without any invariance-aware strategies exhibits comparable performance compared to more advanced and complex DI-GNNs. To inspect where DI-GNNs fall short, we plot in Fig. 4 and Fig. 5 the distribution of edge scores predicted by the detector of `LECI` and `GSAT`, where it is clear that they failed in identifying any sparse invariant subgraph, therefore falling back to a regular-GNN-like behavior. This is especially expected for `SST2-Length`, which is a graph-adapted sentiment classification dataset where node features are contextual embeddings extracted from a BERT-like Transformer (Yuan et al., 2022) – which produces intrinsically globally-correlated node representations by the nature of its self-attention layers (Vaswani et al., 2017). Indeed, BERT-like Transformers are the models achieving state-of-the-art performance in this task (Yuan et al., 2022).

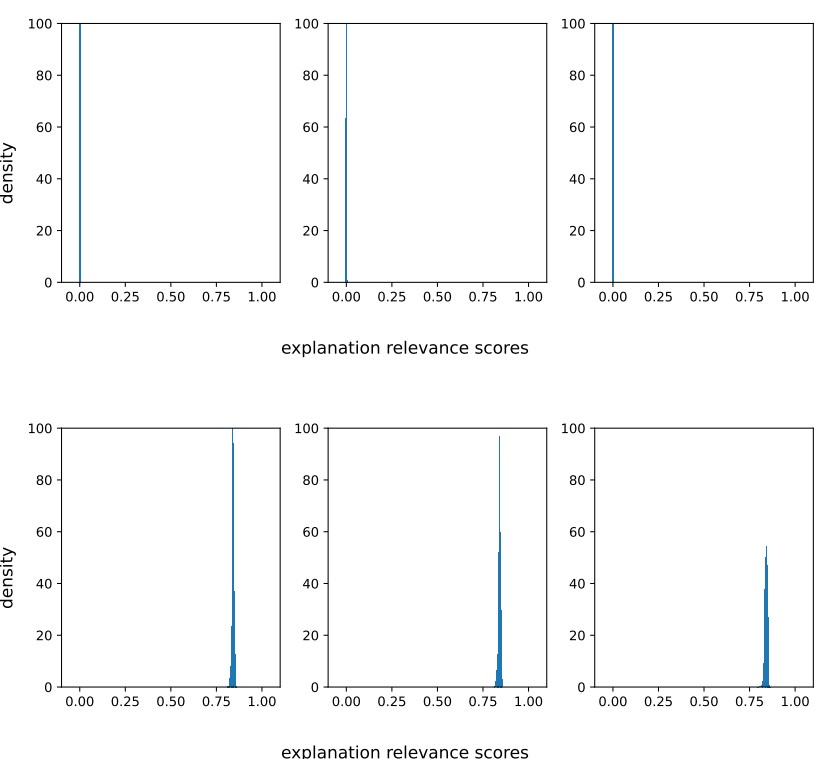

Figure 4: **Histograms of explanation relevance scores for `LECI` (top), and `GSAT` (bottom) on `SST2`** (seed 1). Both models failed in identifying a sparse input subgraph, assigning constant scores (or very close thereof) to every edge.

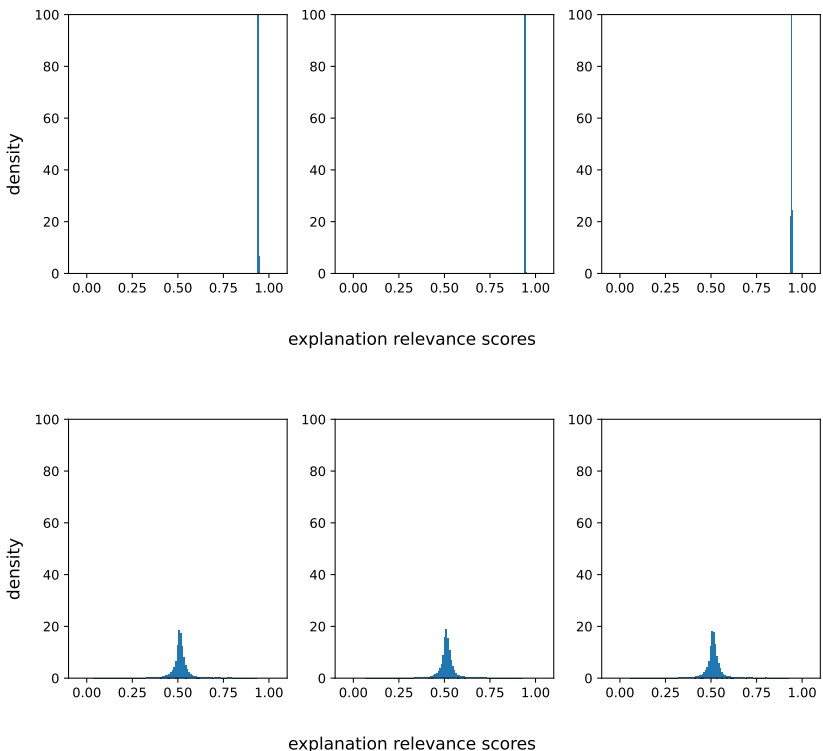

Figure 5: **Histograms of explanation relevance scores for `LECI` (top), and `GSAT` (bottom) on `LBAPcore`** (seed 1). Both models failed in identifying a sparse input subgraph, assigning constant scores (or very close thereof) to every edge.

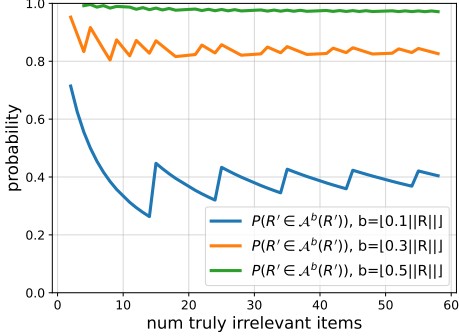

Figure 6: **Probability of deleting at least one truly relevant edge is independent of the number of irrelevant edges if the deletion budget depends on the explanation size.** Given an explanation $R$ with $r$ truly relevant edges ($r = 5$), and a budget b proportional to the size of the explanation, the plot shows $P(R' \in \mathcal{A}_R^b(R'))$ where $R' \sim p_C^b(G)$, for a growing number of irrelevant edges in $R$. The plot shows that the probability is approximately constant, i.e., it does not depend on the number of irrelevant edges. The segments with decreasing behaviour (especially visible for a $10\%$ budget, the blue curve) correspond to areas where the budget is indeed constant, and thus not proportional to the explanation size. For instance, between 1 and 14 irrelevant edges, a budget of $10\%$ corresponds to deleting one edge.

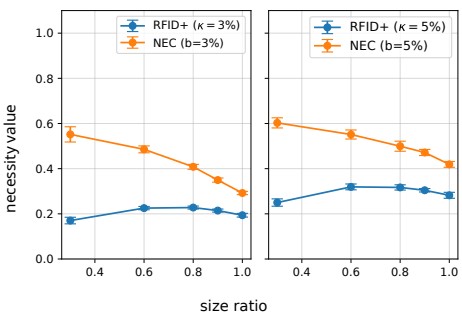

Figure 7: **Dependency of the necessity metrics on the size of the explanation.** RFid+ and Nec of explanations output by GSAT on Motif2-Basis (averaged over 5 seeds) for different explanation sizes (x-axis) and metric hyperparameter $\kappa, b \in \{3\%, 5\%\}$. RFid+ assigns similar or even higher scores to larger explanations, while Nec (with a budget $b$ proportional to the average graph size $\bar{m}$) decreases for increasing explanation size, as expected.

### D.2 SENSITIVITY OF RFid+ AND Nec TO THE EXPLANATION SIZE

As in the proof of Proposition 3, let $\mathcal{S}_R^b$ be the set of subgraphs of $G$ obtained by deleting $b$ edges from $R$ while keeping $C$ fixed, and $\mathcal{A}_R^b = \{G' \in \mathcal{S}_R^b : \Delta(G, G') \geq \epsilon\}$ those subgraphs that lead to a large enough change in $\Delta$. Then, in Fig. 6 we numerically simulate the probability of deleting at least one truly relevant edge from an explanation with a growing number of irrelevant ones, where the number of deletions is proportional to the explanation size. The figure shows that the probability is basically insensitive to the number of irrelevant edges.

In Fig. 7 we integrate the results reported in Table 3 and provide a more detailed experiment showing that for GSAT, RFid+ is in fact insensible to the number of irrelevant edges in the explanation, while Nec with a suitable $p_C$ is not. In Fig. 8 and Fig. 9 we further show that this behavior is in fact general, and occurs across different models and datasets. Specifically, for the same setting delineated in Section 3.2, we report the results also for the LECI (Gui et al., 2023) model over LBAPcore-Assay (Gui et al., 2022) molecular benchmark.

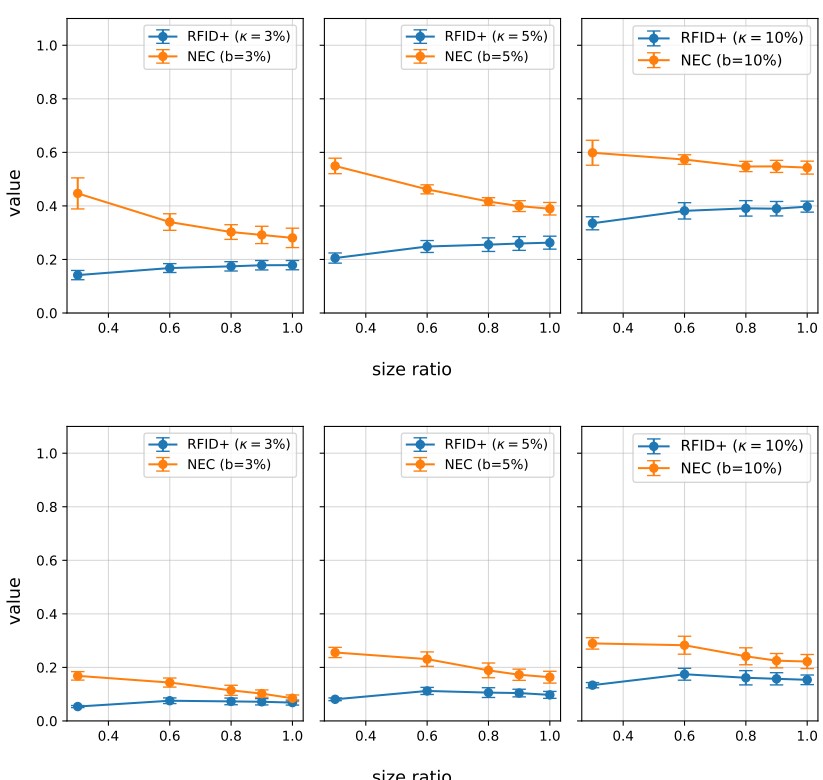

Figure 8: RFid+ **is insensitive to irrelevant edges**. RFid+ and Nec with $p_C^b$ are computed for `LECI` and averaged across 5 seeds on `Motif2-Basis` (top) and `LBAPcore` (bottom) for different explanation sizes (x-axis) and for different metric hyper-parameter $\kappa, b \in \{3\%, 5\%, 10\%\}$. RFid+ assigns similar or even higher scores to larger explanations, while Nec tends to penalize larger explanations.

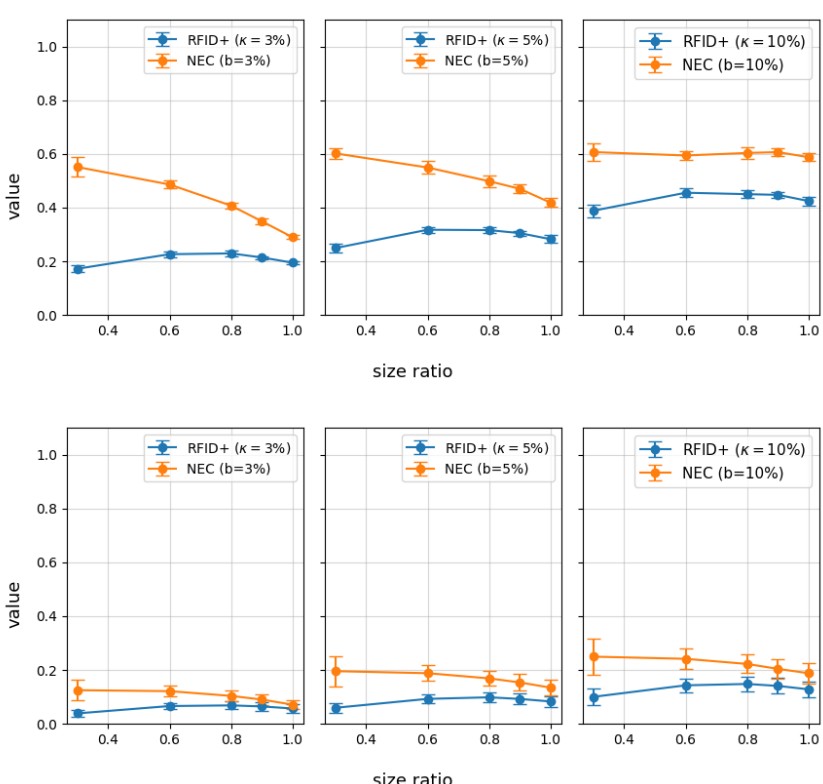

Figure 9: RFid+ **is insensitive to irrelevant edges**. RFid+ and Nec with $p_C^b$ are computed for `GSAT` and averaged across 5 seeds on `Motif2-Basis` (top) and `LBAPcore` (bottom) for different explanation sizes (x-axis) and for different metric hyper-parameter $\kappa, b \in \{3\%, 5\%, 10\%\}$. RFid+ assigns similar or even higher scores to larger explanations, while Nec tends to penalize larger explanations.

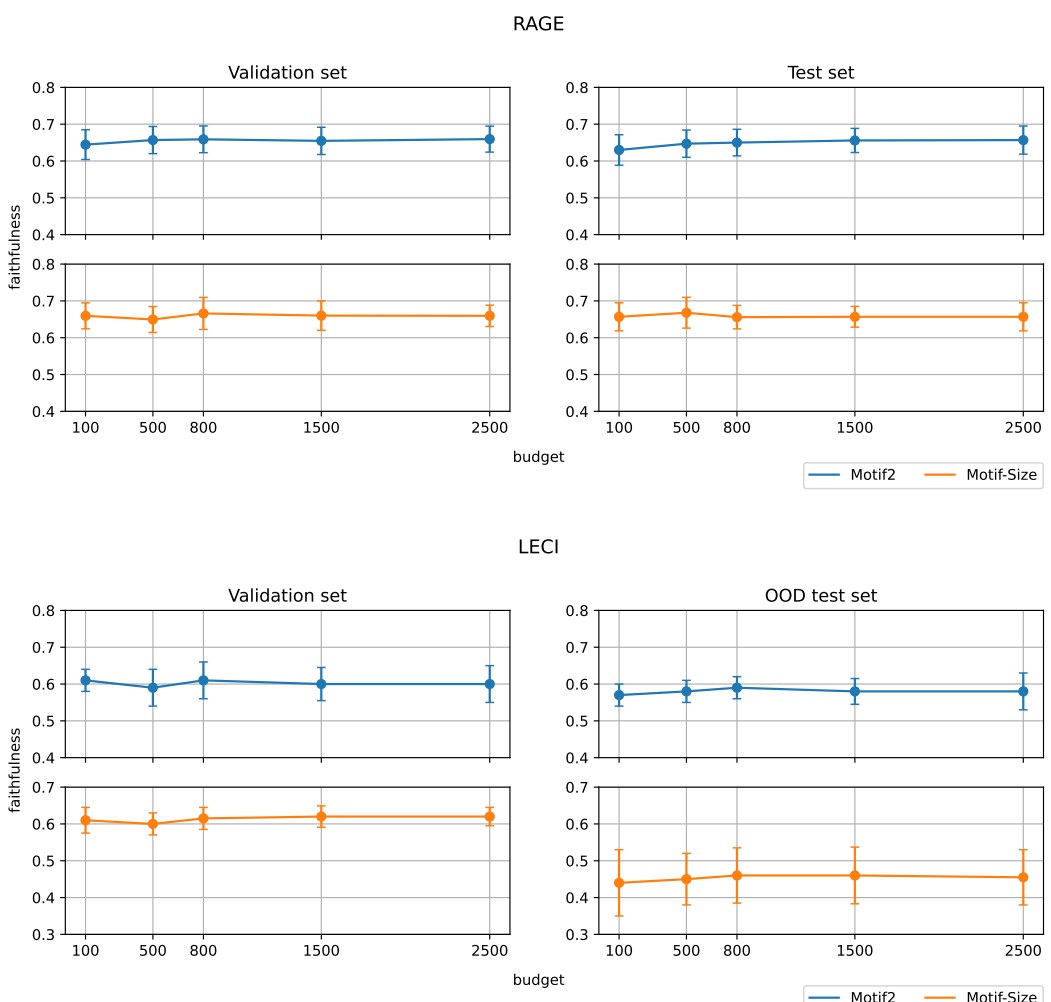

Figure 10: **Ablation study of faithfulness scores** across varying number of number of samples $q_2$.

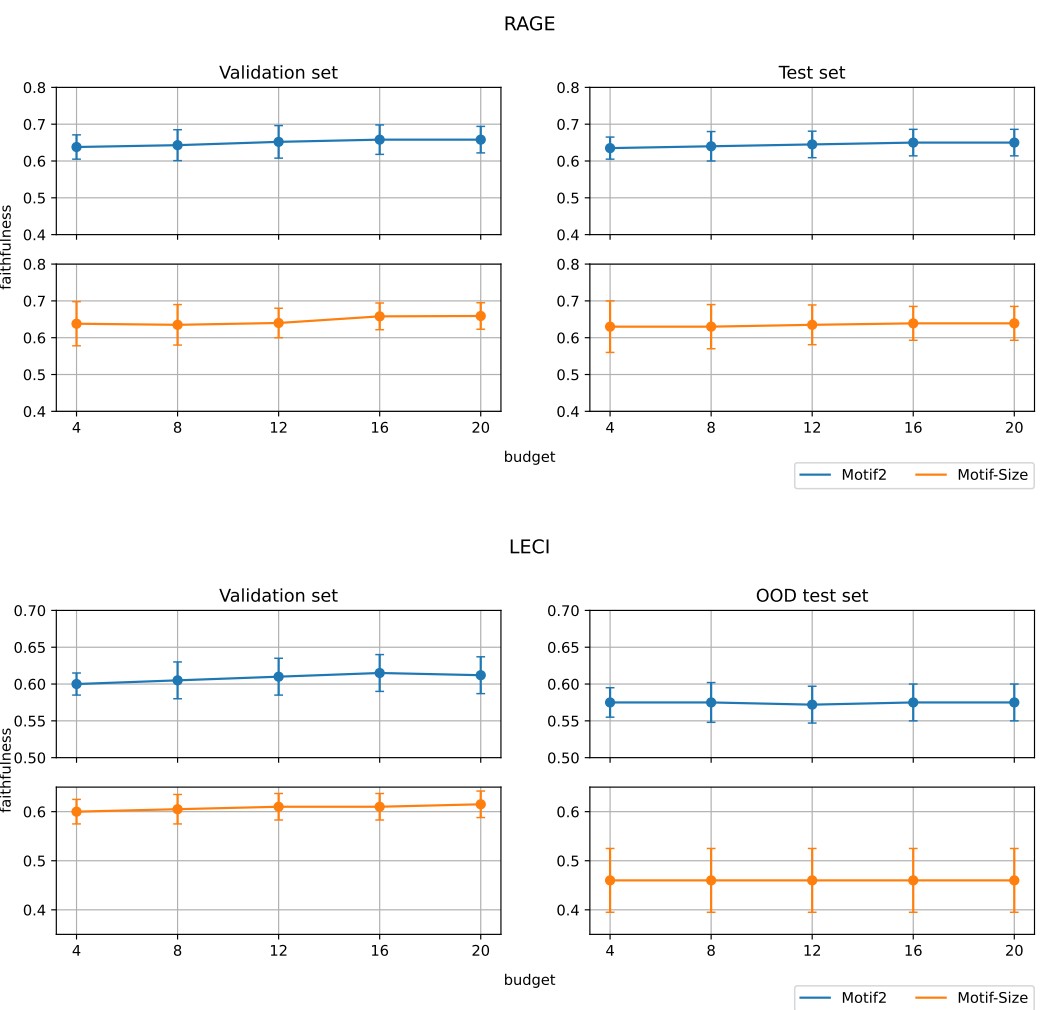

Figure 11: **Ablation study of faithfulness scores** across varying number of interventions $q_1$.

