# OpenReview forum: "Reconsidering Faithfulness in Regular, Self-Explainable and Domain Invariant GNNs"
_ICLR.cc/2025/Conference — ICLR 2025 Poster_

### Official Review · Reviewer_feco · 2024-10-24

**Soundness:** 2
**Presentation:** 3
**Contribution:** 3
**Rating:** 6
**Confidence:** 3

**Summary:**

In this paper, the authors study the faithfulness of explanations generated concerning Graph Neural Networks (GNNs). Specifically, they claim the following main contributions:

1. They claim that current faithfulness metrics are *not interchangeable* and cannot evaluate the explanation quality precisely. These differences can cause the explanations to ignore important elements.
2. They prove that perfectly faithful explanations may be completely uninformative in injective GNNs.
3. They explore the connections between faithfulness and domain invariance.

**Strengths:**

1. GNN explanations' faithfulness is crucial in real-world applications and academic research.

2. The experiments on four popular datasets including Motif2 show an empirical evaluation of the proposed method.

**Weaknesses:**

> W1. Concerning contribution 1:

They claim that current faithfulness metrics are **not interchangeable** and cannot evaluate the explanation quality precisely. In other words, the faithfulness performance of an explanation can vary concerning different metrics (Proposition 1 and Sec. 3.1).

However, "faithfulness" naturally can be evaluated by various metrics, and it is **unnecessary** that the evaluation results should be consistent with each other.

Take the ranking similarity metrics as an example: top-k intersection (precision at k), cosine similarity, NDCG, and other metrics can lead to different results. E.g., ranking A is more similar to ranking B by top-k, while ranking C is more similar to ranking B by cosine similarity, and the divergence among metric results can vary as large as possible. Similar cases can be found in prediction performance cases.


> W2: Concerning contribution 2:

They prove that perfectly faithful explanations may be completely uninformative in injective GNNs (Prop. 4).

However, the details are missing and hard to follow. Specifically, which faithfulness metric is adopted, and what does "match" mean in "$p_r$ matches $N_L(u)$" and "$p_R$ matches $G_A$"?


> W3. Concerning the target tasks:

Are the graph- or node-level classification tasks considered? Definition 1 will be strict for graph-level tasks, especially when the "output" is the distributions rather than categories.

> W4. Concerning the experiments:

The purpose of some experimental results is unclear.

One of the contributions is the new faithfulness metric (as well as Nec and Suf), while their superiority is not explored by further experiments.

Instead, in Table 3 and Table 4, the authors evaluate GSAT (and other backbone models) with/without augmentations by Acc and faithfulness. These models may achieve better or worse performance while their connections to the paper's key contribution are unclear. Besides, the relationship between Acc and faithfulness is underexplored.

In other words, the paper proposes to "reconsider" the faithfulness metric, while comprehensive experiments of comparing metrics are missing.

> W5. Concerning the writing:

W5a. Most figures and tables are really far away from their corresponding statements, requiring to jump over the pages repeatedly.

W5b. Some key statements are missing, e.g., the choice of d in Lines 279, and what does "value" mean in the y-axis?

W5c. Are the graph- or node-level classification tasks considered? Or do different propositions/statements fit different tasks?

**Questions:**

Please address the concerns and questions in the *weakness* part.

---

> ### Author Response · Authors · 2024-11-18
> **Response to Reviewer feco**
>
> Thank you for your feedback. Please find below our answers to your questions.
>
> ---
>
> > **W1**
>
> We completely agree. We do not claim that all faithfulness metrics should be consistent with each other, but we do point out that existing ones do not and that, therefore, care should be taken when choosing and communicating faithfulness results.
>
> Indeed, **differences between ranking metrics, just like for others, are well-documented**, and oftentimes multiple metrics are reported to show the generalizability of the results [1,2,3]. A similar argument applies to Accuracy and F1 score, where one is chosen over the other based on insights about the class distribution.
>
> **For faithfulness**, to the best of our knowledge, **such differences have not been documented**, and in fact, previous works typically choose faithfulness metrics without motivating their choice and considering the distribution of interventions more like an implementation detail rather than a key factor of the metric being evaluated [4,5,6]. For instance, we showed that some necessity metrics do not properly account for the number of irrelevant edges in the explanation (Prop. 2-3), therefore encoding different semantics than expected.
>
> For the sake of completeness, we added a discussion on how to choose the most appropriate metric configuration depending on the task at hand in the revised Appx. C.4.
>
> ---
>
> > **W2: Details of Prop. 4 are missing**
>
> Thank you for pointing this out.  Prop. 4 works for any choice of $p_C$ and $p_R$ subsampling the input graph, just like the ones present in Table 1. This is because any perturbation sampled from them will inevitably lead to a change in the model's output, due to our injectivity assumption.  As for $d$, the statement holds for any divergence between output probability distributions, and $d$ being the difference in the prediction likelihood. **We updated the statement and proof of Prop. 4 accordingly**.
>
> Prop. 4, therefore, holds for any faithfulness metric fitting this template.
>
> ---
>
> > **W2: What does "$p_R$ matches $N_L(u)$" mean in  Prop. 4?**
>
> We believe this is a slight misunderstanding. In fact, "match" was referred to "$R_A$", and not to "$p_R$". Therefore, "matches" simply means the explanation’s topology should be equal to either $N_L(u)$ or $G_A$, which in both cases is the computational graph of the prediction.
>
> We rephrased Prop. 4 to remove this ambiguity.
>
> ---
>
> > **W3 & W5c**
>
> Thank you for pointing this out.  **Our results hold for both node and graph classification**. Recalling that given a node $u$, modular GNNs would output an explanation $R_A \subseteq N_L(u)$, our results can be interpreted as follows:
> - **Def 1**, **Def 2**, and **Prop 1** are expressed in terms of an input graph $G_A$, but the definitions are general and hold also for node-level tasks. In this setting, $C_A = N_L(u) \setminus R_A$. Then, the definitions apply verbatim.
> - **Prop. 2 and 3** focus on the explanation $R_A$, thus apply directly to node-level tasks. In fact, the example of Figure 1 is about a node classification task as it offers an easier intuition.
> - **Prop 4** differs based on the task. This is already specified in the statement.
> - **Prop 5** and **Thm. 1** hold for both node and graph tasks, provided they respect the domain-invariance assumptions listed in Section 5.
>
> We clarified this in line 160 of the revised PDF.
>
> ---
>
> > **W4: The superiority of the new metric is not explored**
>
> We claim that some **necessity** metrics are better than others: we have shown both theoretically (Prop. 2 and Example 1) and empirically (Fig 2 and Appx. D.2) that our variant of $p_C^b$ is better at penalizing overly large explanations.
>
> We did not claim superiority of specific **sufficiency** metrics.  What we argue is that different choices of $p_R$ can lead to radically different faithfulness semantics, so if one does not clarify what $p_R$ they used, faithfulness results can be easily misinterpreted.
>
> ---
>
> >  **W4: The authors report Acc and faithfulness. Their connections to the paper's key contribution are unclear. The relationship between Acc and faithfulness is underexplored.**
>
> Table 3 and Table 4 are connected with our second main contribution, which in addition to showing the impossibility of faithfulness for injective GNNs, aims to investigate the practical relationship between some architectural desiderata for modular GNNs and faithfulness (lines 20-23). We include Acc as a sanity check to assess whether faithfulness strategies consistently affect Acc. A formal link between OOD Acc and Faithfulness is then investigated in Section 5, with also a measure of their correlation in lines 494-495.
>
> We updated the revised PDF in line 363 clarifying this.
>
> ---
>
> > **W5a**
>
> Good point. We have rearranged them.
>
> > **W5b**
>
> These details were available in Appx. B.2.2. We added them in line 264 and updated the y-axis to indicate that it refers to necessity computed by the two metrics.

---

> > ### Author Response · Authors · 2024-11-18
> > **Adding the References to the Previous Authors' Comment**
> >
> > [1] DeepRank: A New Deep Architecture for Relevance Ranking in Information Retrieval. 2017
> >
> > [2] PiRank: Scalable Learning To Rank via Differentiable Sorting. 2021
> >
> > [3] Combining variable neighborhood with gradient ascent for learning to rank problem. 2023
> >
> > [4] Graphframex: Towards systematic evaluation of explainability methods for graph neural
> > networks. 2022
> >
> > [5] Evaluating Explainability for Graph Neural Networks. 2023
> >
> > [6] Explaining the Explainers in Graph Neural Networks: a Comparative Study. 2024

---

> > ### Comment · Reviewer_feco · 2024-11-22
> >
> > Thanks for further clarifying the misunderstandings and confusion.
> >
> > > RW1:
> >
> > a) The key concern is about the motivation/goal of the paper. As emphasized in the abstract (Lines 16-18) and Introduction (Lines 47-50), the inconsistency among different faithfulness metrics is posed as one key contribution. However, it is contrastive to the reply
> > "We do not claim that all faithfulness metrics should be consistent with each other".
> >
> > b) Not sure what you mean by "documented". It is a trend that XAI works [1,2] evaluate the "faithfulness" by many, instead of one, metrics.
> >
> > > RW4
> >
> > Again, the paper claims to "reconsider" the faithfulness metrics, however, the most experimental results (e.g., Table 3, 4, and Figure 3) are quite far away from the topic. Even more, the original Figure 2 does not include the necessary information to disclose the desired messages.
> >
> > It is worth noticing that reviewers (as well as other readers) are **NOT** required to read the appendix, and the main manuscript is supposed to be **self-contained**.
> >
> > [1] Chrysostomou, George, and Nikolaos Aletras. "Improving the faithfulness of attention-based explanations with task-specific information for text classification." ACL 2021.
> > [2] DeYoung, Jay, et al. "ERASER: A benchmark to evaluate rationalized NLP models." ACL 2020.

---

> ### Author Response · Authors · 2024-11-22
> **Follow-up to Reviewer feco**
>
> Thank you for your feedback.
>
> ---
>
> > **RW1.a)**
>
> We apologize for any confusion.  We have revised our manuscript to clarify that our criticism is not so much about interchangeability per se, as we agreed that evaluation metrics can naturally disagree, but rather about how these metrics are used. Our main concern is that GNN developers or certification bodies might use faithfulness metrics without much thought, assuming they *are* interchangeable, while they are not.  In practice, this can lead to unreliable or biased rankings of proposed architectures and their explanations.
>
> To make this more concrete, we have compared empirically how changing the distribution $p_R$ computing the sufficiency alters the absolute values of SUF significantly and also the ranking of GNN models in practice.
>
> In particular, we tested four different distributions:
> - $p_R^{id_1}$ and $p_R^{ood_1}$ replace the complement $C_A = G_A \setminus R_A$ of an input graph with that of another sample $G_A’$, taken respectively from the ID or OOD data split.
> - $p_R^{id_2}$ and $p_R^{ood_2}$ subsample the complement of each graph by randomly removing a fixed budget of edges. The metric is computed either on ID or OOD samples, respectively.
>
> The table below shows for each distribution the ranking of models and the value of SUF between parentheses.
>
> | Model | \| | Motif-size | | \| | | Motif2 |  |
> | :----------: | ----------- | ----------- | ---------- | :----------: |:----------:  | :----------: | :----------: |
> |                  | \| | $p_R^{id_1}$   | $p_R^{id_2}$  | \|  |  | $p_R^{id_1}$ | $p_R^{id_2}$ |
> | LECI         | \| | 2 (79)         | 2 (83)        | \| |  | 1 (81) | 2 (82) |
> | GSAT        | \| | 1 (83)         | 1 (86)       | \| |  | 2 (78) | 1 (84) |
> | CIGA         | \| | 3 (70)         | 3 (72)       | \| |  | 3 (65) | 3 (73) |
> |                   | \| | $p_R^{ood_1}$  | $p_R^{ood_2}$  | \| |  | $p_R^{ood_1}$ | $p_R^{ood_2}$ |
> | LECI          | \| | 1 (81)         | 1 (89)       | \| | | 2 (83) | 1 (88) |
> | GSAT        | \| | 2 (73)         | 2 (80)        | \| |  | 3 (76) | 3 (79) |
> | CIGA         | \| | 3 (55)         | 3 (70)        |  \| | | 1 (85) | 2 (86) |
>
>
> In two cases out of four, just changing the type of perturbation allowed by $p_R$ (replacing the complement vs removing random edges; across columns) changes the ranking of models. Instead, changing the graph distribution (ID vs OOD; across rows) alters the ranking in every case.
>
> We will *replace the example in Figure 1 of the PDF* with this table by tomorrow.
>
> ---
>
> > **RW1.b)**
>
> We focus on GNNs, where this trend does not apply.  Besides papers that propose new metrics (like the ones we reference in the text), papers that propose new methods or survey papers generally tend to stick to a single definition  [5,6,7,8,9].  We agree the trend you point out is good and needed, and our work can be viewed as an attempt to start this trend also for GNNs.
>
> [7] Interpreting Graph Neural Networks for NLP with Differentiable Edge Masking. 2022
>
> [8] A Survey on Explainability of Graph Neural Networks. 2023
>
> [9] A Comprehensive Survey on Trustworthy Graph Neural Networks: Privacy, Robustness, Fairness, and Explainability. 2024
>
> ---
>
> > **RW4)**
>
> Please note that we address “Reconsidering Faithfulness”, and not “Reconsidering Faithfulness Metrics”.  What we mean by this is that we aim at reconsidering faithfulness from a broader perspective, and not just in the way it is computed.  In fact, the paper is organised into three main contributions, each touching different aspects of faithfulness:
> - Reconsidering how faithfulness metrics are implemented (Section 3)
> - Reconsidering how GNN architectures aim to achieve faithful explanations (Section 4)
> - Reconsidering the role of faithfulness outside of the XAI community, and in particular its role in the Domain Invariance literature (Section 5)
>
> We believe that under this lens those three sections and all the experiments fit the overall message of “Reconsidering Faithfulness”.  We will update the Introduction by adding the following sentence at the end:
>
> "*Overall, the previous three main contributions move towards reconsidering faithfulness in a broad sense, i.e., by i) reconsidering how it is computed, ii) reconsidering how GNN architectures aim at achieving it, and iii) reconsidering its role outside of the XAI literature.*"
>
> ---
>
> > **The key message of Fig 2 is not clear**
>
> Thank you for the feedback. We will streamline the discussion of Figure 2 in the main text, strengthening the link between our theoretical analysis of necessity metrics and Figure 2. Also, as the picture can be difficult to grasp, we will replace it with a textual description capturing its core idea, along with all the details needed to understand the key takeaways of this empirical analysis.
>
> We will update the PDF by tomorrow.
>
> ---
>
> We appreciate your valuable feedback and we'll remain available for any further clarification.

---

> > ### Comment · Reviewer_feco · 2024-11-25
> >
> > Thanks for replying to the questions and I have no further concerns.
> >
> > I will raise the score and best luck to your submission.

---

### Official Review · Reviewer_5LA6 · 2024-10-29

**Soundness:** 3
**Presentation:** 2
**Contribution:** 3
**Rating:** 6
**Confidence:** 3

**Summary:**

This paper studies the faithfulness of explanations in modular GNNs. It addresses several problems in explanation metrics, and claims through analysis: 1) not all metrics are the same; 2) faithfulness metrics are not interchangeable; not all metrics are equally reliable; 3) for regular injective GNNs, strict faithful explanations are trivial. Finally, it claims faithfulness is key to OOD generalization. Extensive experiments justify the claims in the paper.

**Strengths:**

1. The paper show that a) existing metrics are not interchangeable. b) optimizing for faithfulness is not always a sensible design goal. c). faithfulness is tightly linked to out-of-distribution generalization. The novelty is good.
2. The paper present an in-dept analysis of existing graph XAI metrics.
3. The experiments are solid.

**Weaknesses:**

1. After reading the paper, I am in question of what metrics are suggested for a GNN explanation task. Considering a normal setting, a post-hoc explainer for a trained to-be-explained GNN, how to choose the metric? Can you draw a clear conclusion throughout the analysis in the paper? Therefore, I question the significance of the paper. To improve the paper, I highly suggest the authors include a section summarizing key takeaways and practical guidance on selecting appropriate faithfulness metrics for various GNN explanation settings, tailored to different types of models (e.g., modular GNNs).
2. In section5, how to determine if the explanation is not strictly sufficient? It would be beneficial to improve the paper by providing a concrete example or procedure for determining if an explanation fails to be strictly sufficient.

**Questions:**

See the weaknesses

---

> ### Author Response · Authors · 2024-11-18
> **Response to Reviewer 5LA6**
>
> We thank the reviewer for their feedback.  We are glad they found our contribution novel and supported by an in-depth analysis and solid experiments.  Below we address their remarks.
>
> ---
>
> > **I highly suggest the authors include a section summarizing key takeaways and practical guidance on selecting appropriate faithfulness metrics for various GNN explanation settings, tailored to different types of models**
>
> We have added a discussion of what choosing different interventional distributions ($p_R$ and $p_C$) and divergences ($d$) might entail in Appendix C.4 of the updated PDF.  In short, some choices yield “stricter” faithfulness metrics and others “looser” metrics, and there is a monotonic relationship between these.
>
> To see this, consider two different choices for $p_R$:
>
> - *Strictest choice*: $p^1_R$ allows to modify the complement of the explanation in every conceivable way via node/edge addition and removal and arbitrary feature perturbations. Being strictly sufficient wrt $p^1_R$ means being invariant to any possible change to the complement.
>
> - *Loosest choice*: $p^2_R$ allows only node/edge removal. This is the most narrow distribution, and it corresponds to most metrics in Table 1.
>
> Similar arguments hold for $p_C$ and $d$, see Appendix C.4.
>
> Clearly, there is a monotonic relationship between metrics: achieving high sufficiency (resp. faithfulness) wrt a stricter metric entails high sufficiency (resp. faithfulness) wrt looser metrics.
>
> The choice of best metric is however application dependent.  Stricter metrics are tighter and make sense in high-stakes applications, where faithfulness is a must, and for providing adversarial robustness guarantees (for, e.g., Machine Learning as a Service), while looser metrics are more meaningful if adversarial inputs are not a concern.
>
> These insights apply to both post-hoc and modular settings.
>
> Our key point is however that some metrics are surprisingly **flawed**, as we show for common necessity metrics in Section 3.2.
>
>
> ---
>
> > **How to determine if the explanation is not strictly sufficient?  It would be beneficial to improve the paper by providing a concrete example or procedure for determining if an explanation fails to be strictly sufficient**
>
> Testing if an explanation is not strictly sufficient amounts to finding a counterexample showing that a modification outside of the explanation leads to a large enough prediction change. This can be done naively with Monte Carlo sampling from $p_R$, or by devising more efficient algorithms inspired to, for example, finding adversarial examples [1].
>
> We added in Appendix C.3 a discussion regarding testing for non-strict sufficiency, along with the sketch of an algorithm.
>
> [1] Adversarial training for graph neural networks: Pitfalls, solutions, and new directions. NeurIPS 2024.

---

> > ### Author Response · Authors · 2024-11-21
> > **Adding New Empirical Evidence in Support**
> >
> > Dear reviewer,
> >
> > We run some additional experiments to complement our answer regarding the practical implications of the following statement:
> >
> > > *Faithfulness metrics are not interchangeable, in the sense that explanations that are highly faithful according to one metric can be arbitrarily unfaithful for the others (lines 185-187)*
> >
> > In the following table, we show that by simply changing the underlying distribution $p_R$ computing the sufficiency, not only can the absolute values significantly change, but the ranking of models can also change.
> >
> > In particular, we tested four different distributions:
> >
> > - $p_R^{id_1}$ and $p_R^{ood_1}$ replace the complement $C_A = G_A \setminus R_A$ of an input graph with that of another sample $G_A’$, taken respectively from the ID or OOD data split.
> > - $p_R^{id_2}$ and $p_R^{ood_2}$ subsample the complement of each graph by randomly removing a fixed budget of edges. The metric is computed either on ID or OOD samples, respectively.
> >
> >
> > | # Ranking (SUF) | \| | Motif-size | | \| | | Motif2 |  |
> > | :----------: | ----------- | ----------- | ---------- | :----------: |:----------:  | :----------: | :----------: |
> > |                  | \| | $p_R^{id_1}$   | $p_R^{id_2}$  | \|  |  | $p_R^{id_1}$ | $p_R^{id_2}$ |
> > | LECI         | \| | 2 (79)         | 2 (83)        | \| |  | 1 (81) | 2 (82) |
> > | GSAT        | \| | 1 (83)         | 1 (86)       | \| |  | 2 (78) | 1 (84) |
> > | CIGA         | \| | 3 (70)         | 3 (72)       | \| |  | 3 (65) | 3 (73) |
> > |                   | \| | $p_R^{ood_1}$  | $p_R^{ood_2}$  | \| |  | $p_R^{ood_1}$ | $p_R^{ood_2}$ |
> > | LECI          | \| | 1 (81)         | 1 (89)       | \| | | 2 (83) | 1 (88) |
> > | GSAT        | \| | 2 (73)         | 2 (80)        | \| |  | 3 (76) | 3 (79) |
> > | CIGA         | \| | 3 (55)         | 3 (70)        |  \| | | 1 (85) | 2 (86) |
> >
> >
> >
> >
> >
> > In two cases out of four, just changing the type of perturbation allowed by $p_R$ (replacing the complement vs removing random edges; across columns) changes the ranking of models. Instead, changing the graph distribution (ID vs OOD; across rows) alters the ranking in every case.
> >
> > Appendix C. 5 now includes the table above, with further details and the results for CMNIST.

---

> > > ### Comment · Reviewer_5LA6 · 2024-11-25
> > >
> > > I thank the authors have answered all of my questions, and I will keep my score.

---

### Official Review · Reviewer_s1hn · 2024-10-29

**Soundness:** 3
**Presentation:** 3
**Contribution:** 3
**Rating:** 6
**Confidence:** 4

**Summary:**

Faithfulness has been one of the widely used metrics in explainable artificial intelligence research. Intuitively, an explanation is said to be faithful if it accurately reflects the true reasoning process of the underlying model. However, the investigation of faithfulness with respect to graph neural networks has been limited to empirical analysis of generated explanations and proposing new metrics. In this work, the authors highlight the difference between existing faithfulness metrics and argue about the trade-off between them, i.e., existing faithfulness metrics are at odds with each other. Further, the authors show that optimizing for faithfulness is not ideal and leads to a trade-off between the expressiveness of the model and usefulness of faithful explanations.

**Strengths:**

1. The paper is well-written and the motivation is very clear.

2. The authors detail the importance of the faithfulness metric in graph explanations, highlighting the pitfalls of diverse metrics and the trade-offs between them.

3. The paper presents an interesting analysis between regular, self-explainable, and domain-invariant graph neural networks.

**Weaknesses:**

1. The relation between faithfulness and out-of-distribution performance is unclear. The statement "faithfulness is tightly linked to out-of-distribution generalization" is weak and doesn't highlight the exact implication of faithfulness on OoD performance. While the authors show the analysis in Section 5, it would be beneficial to the readers if the authors could provide a brief explanation describing this relation in simple terms.

2. The paper lacks a clear description and implication of the theoretical analysis. Formally, each theorem statement is followed by a proof sketch, which describes the main techniques used to derive the lower/upper bound and the practical implications of the bounds with respect to the GNN, explanation, and faithfulness. Addressing this will make the theoretical analysis of the paper much stronger.

**Questions:**

Please look at the weaknesses for open questions.

---

> ### Author Response · Authors · 2024-11-18
> **Response to Reviewer s1hn**
>
> We thank the reviewer for their feedback.  We are glad they found our analysis well motivated and interesting, and useful for identifying pitfalls of faithfulness research.  Below we address their remarks.
>
> ---
>
> > **The relation between faithfulness and out-of-distribution performance is unclear.  It would be beneficial if the authors could provide a brief explanation describing this relation in simple terms.**
>
> Thank you for the suggestion.  We have updated the *Faithfulness is key to OOD generalization* paragraph (lines 64+) to provide a clearer intuition, as follows:
>
> “*Specifically, prior work tackles domain invariance by constructing DIGNNs that isolate the domain invariant portion of the input and use it for prediction. **We show that extracting a domain invariant subgraph is not enough for a truly domain invariant GNN: unless the subgraph is also faithful, the information from the domain-dependent components of the input can still influence the prediction, thus preventing domain invariance**. This highlights a key limitation in current design and evaluation strategies for DIGNNs, which neglect faithfulness altogether.*”
>
> We had to postpone the actual analysis to Section 5 due to space constraints.
>
> ---
>
> > **The paper lacks a clear description and implication of the theoretical analysis.**
>
> Our theoretical analysis is structured as follows:
>
> - **Prop. 1** shows that faithfulness metrics are not interchangeable, meaning that explanations may be faithful wrt to one metric but arbitrarily unfaithful wrt to others. The implication is that faithfulness values must be properly contextualized wrt the parameters $p_C$, $p_R$, and $d$ to avoid misinterpretations (lines 185-196).
>
> - **Prop. 2 and 3** formalize the intuition that different *necessity* metrics have different semantics. We provide intuition for each proof right before each statement. Their practical implications are explored in Section 3.2. In short, we illustrate how to devise a necessity metric that properly accounts for the number of truly irrelevant items (wrongly) covered by an explanation.
>
> - **Prop. 4** shows that for injective regular GNNs, the explanation must cover the entire computational graph. This result implies that faithful explanations do not depend on model weights, and are thus uninformative (line 315)
>
> - **Prop. 5** shows that even if a modular GNN correctly identifies invariant subgraphs, unless the subgraph is strictly sufficient, the model as a whole cannot be domain invariant. This result demonstrates that subgraph invariance alone is not enough for truly invariant DI-GNNs (line 454-460), contrary to what is commonly assumed in the domain invariance literature.
>
> - **Theorem 1** takes a step further and shows that the degree of sufficiency and plausibility upper bound the change in prediction likelihood – a proxy to measure the OOD generalisation ability of a GNN. The implication of this result is that we can devise better DI-GNNs by also considering sufficiency when designing and evaluating them (lines 485-489).
>
> We updated Section 7 in the uploaded revised PDF to more clearly convey the main takeaways, highlighting for each of them how they are connected to the theoretical analysis.
>
> ---
>
> > **Formally, each theorem statement is followed by a proof sketch, which describes the main techniques used to derive the lower/upper bound and the practical implications of the bounds**
>
> Unfortunately, we cannot include a proof sketch for each result, due to space constraints.
>
> However, following your suggestion, we have updated the text in lines 483-484 to briefly illustrate the main techniques and assumptions used to prove Theorem 1, as follows:
>
> “*The Theorem is proved by applying the triangular inequality and basic properties of the expectation and relies on two main assumptions: the domain invariance of $R_A^\*$, and the Lipschitzness of GNNs.*”
>
> As we mentioned, the practical implications are discussed in lines 485-489.

---

> > ### Comment · Reviewer_s1hn · 2024-11-22
> >
> > Thank you for providing the clarifications. I will wait for the response from other reviewers and increase my rating accordingly. Best of luck with the submission!

---

> ### Author Response · Authors · 2024-12-01
> **Final Answer to Reviewer Comment**
>
> We are glad to hear that our comments clarified your concerns.  Given that all other reviewers have now replied, and that the discussion period will end in about two days, we kindly ask whether you would consider reflecting your improved opinion in an increased score.  Please let us know if there is anything else you wish us to clarify.
>
>
> Regardless, thank you for your time and valuable feedback,
>
> The Authors

---

> > ### Comment · Reviewer_s1hn · 2024-12-02
> >
> > Thank you for the reminder, Authors. Looking at all the responses from other reviewers, I would like to maintain my score of weak acceptance.

---

### Official Review · Reviewer_2N4z · 2024-11-03

**Soundness:** 2
**Presentation:** 2
**Contribution:** 2
**Rating:** 6
**Confidence:** 2

**Summary:**

This paper addresses a timely and impactful topic by exploring the limitations of current metrics used to evaluate the faithfulness of explanations in graph neural networks (GNNs). The authors make three primary contributions: they highlight that existing metrics are not interchangeable, challenge the assumption that faithfulness should always be a design goal, and link faithfulness to out-of-distribution generalization.

**Strengths:**

This paper addresses a timely and impactful topic by exploring the limitations of current metrics used to evaluate the faithfulness of explanations in GNNs. The paper presents a range of experiments to support these claims, adding depth and rigor to the discussion.

**Weaknesses:**

1. Clarity in Introduction: The logical flow between sentences in the introduction could be strengthened. For example, the connection between lines 35 and 36 feels disjointed, affecting readability and coherence.

2. Writing Clarity: There are instances of unclear terminology and abbreviations. For instance, the abbreviation “cf.” on line 46 may be confusing. Additionally, on line 77, there’s a mixture of notation, specifically "|G|" and "∥G∥", which could be standardized for consistency.

3. Lack of Proposed Solutions: While the paper highlights significant limitations in existing metrics, it would be more impactful if it suggested potential solutions or proposed a new metric that addresses these limitations. Presenting an alternative would strengthen the contributions and provide practical guidance for future research.

**Questions:**

See in Weaknesses.

---

> ### Author Response · Authors · 2024-11-18
> **Response to Reviewer 2N4z**
>
> We thank the reviewer for their feedback.  We are glad they found our study timely and significant and our analysis rigorous and in-depth.  Below we address their remarks.
>
> ---
>
> > **The logical flow between sentences in the introduction could be strengthened, e.g., lines 35 and 36**
>
> We have polished the Introduction, please see the **updated PDF**.
>
> If you could point us to any other passages that are unclear, we’d be glad to clarify them.
>
> ---
>
> > **The abbreviation “cf.” may be confusing. Additionally, there’s a mixture of notation, specifically "|G|" and "∥G∥", which could be standardized for consistency.**
>
> We replaced “*cf.*” with “*see*” and removed usage of $|G|$ entirely.
>
>
> ---
>
>
> > **While the paper highlights significant limitations in existing metrics, it would be more impactful if it suggested potential solutions or proposed a new metric that addresses these limitations**
>
> Our main goal is to highlight significant but neglected issues with existing faithfulness metrics and their usage, in the same vein as many other impactful critique papers in the XAI literature [1, 2, 3].
>
> However, it is not true that we propose no solutions. In Section 3.2, **we suggest strategies for improving necessity metrics** and show both theoretically (Prop. 2 and Example 1) and empirically (Figure 2 and Appendix D.2) that our proposed $p_C^b$ is better for evaluating unnecessarily large explanations compared to existing alternatives.
>
> [1] Adebayo et al., Sanity Checks for Saliency Maps. NeurIPS 2018.
>
> [2] Rudin, Stop explaining black box machine learning models for high-stakes decisions and use interpretable models instead. Nature Machine Intelligence 2019.
>
> [3] Ghorbani et al., Interpretation of Neural Networks Is Fragile. AAAI 2019.

---

> > ### Author Response · Authors · 2024-11-25
> > **Discussion period ending soon**
> >
> > Dear Reviewer,
> >
> > As the discussion period is about to end, we would like to kindly ask whether your concerns were addressed in our previous comment, or if further clarifications are needed.
> >
> > We'd be happy to follow-up with any further comment.
> >
> > Thank you.
> >
> > The Authors

---

> > ### Comment · Reviewer_2N4z · 2024-11-26
> >
> > Thank you for addressing my questions. I have no further concerns at this time. I will increase the score, and I wish you the best of luck with your submission.

---

### Author Response · Authors · 2024-11-18
**Summary of major changes**

We thank all the reviewers for their feedback. We made available a revised version of our manuscript to support the discussion, where changes are marked in red. In short, our major changes can be summarized as:

- We polished the Introduction to enhance the flow between sentences (rev. 2N4z)
- We linked each key takeaway in our conclusions to the relative theoretical analysis to improve their clarity (rev. s1hn)
- We added two new sections in the Appendix: we describe an algorithm for testing for non-strict sufficiency in Appx. C.3, and we discuss practical considerations regarding the choice of the most appropriate faithfulness metric in Appx. C.4 (rev. 5LA6)
- We clarified that our theoretical analysis holds for both graph and node classification (rev. feco)

Please find below our answers to questions raised by the reviewers.

---

> ### Author Response · Authors · 2024-11-23
> **Summary of the new revision**
>
> Dear Reviewers,
>
> We have uploaded a new revision of the PDF incorporating all the requested further clarifications.  The major changes of this revision can be summarized as:
>
> - We incorporated new experiments in the main text to communicate more concretely the practical implications of "*metrics are not interchangeable*" in the new Table 2
>
> - We compacted the old Figure 1 into the new Table 3, as this allows for a more intuitive understanding of the main message. Figure 1 was however not deleted, but moved to the Appendix for completeness.
>
> We would like to follow up to see if our responses address your concerns or if you have further questions.
> We would really appreciate the opportunity to discuss this further and know whether our response has addressed your concerns.
>
> Thank you again.
>
> The Authors

---

### Meta-Review · Area_Chair_7Z5C · 2024-12-21

**Metareview:**

This is an important and timely contribution to faithfulness in GNNs. The Reviewers are unanimously in support of accepting the paper post-rebuttal, and I believe it will spark important discussions at the conference. A clear accept.

**Additional Comments On Reviewer Discussion:**

Initially the majority of the reviewers were against accepting this paper, but the Authors made a concentrated effort to address the issues raised by the Reviewers, which resulted in score increases throughout. While support is in the "weak" acceptance category across all Reviewers, I do believe that the majority of the raised weaknesses concern clarity in the writing more than technical soundness, and therefore I have no reservations to recommend acceptance in the paper's current form.

---

### Decision · Program_Chairs · 2025-01-22

Accept (Poster)